# FEDERATED LEARNING WITH DYNAMIC CLIENT ARRIVAL AND DEPARTURE: CONVERGENCE AND RAPID ADAPTATION VIA INITIAL MODEL CONSTRUCTION

## ABSTRACT

While most existing federated learning (FL) approaches assume a fixed set of clients in the system, in practice, clients can dynamically leave or join the system depending on their needs or interest in the specific task. This dynamic FL setting introduces several key challenges: (1) the objective function dynamically changes depending on the current set of clients, unlike traditional FL approaches that maintain a static optimization goal; (2) the current global model may not serve as the best initial point for the next FL rounds and could potentially lead to slow adaptation, given the possibility of clients leaving or joining the system. In this paper, we consider a dynamic optimization objective in FL that seeks the optimal model tailored to the currently active set of clients. Building on our probabilistic framework that provides direct insights into how the arrival and departure of different types of clients influence the shifts in optimal points, we establish an upper bound on the optimality gap, accounting for factors such as stochastic gradient noise, local training iterations, non-IIDness of data distribution, and deviations between optimal points caused by dynamic client pattern. We also propose an adaptive initial model construction strategy that employs weighted averaging guided by gradient similarity, prioritizing models trained on clients whose data characteristics align closely with the current one, thereby enhancing adaptability to the current clients. The proposed approach is validated on various datasets and FL algorithms, demonstrating robust performance across diverse client arrival and departure patterns, underscoring its effectiveness in dynamic FL environments.

## 1 INTRODUCTION

Federated learning (FL) is a decentralized machine learning paradigm that facilitates collaborative model training across multiple clients, such as smartphones and Internet of Things (IoT) clients, without exchanging individual data. Instead of transmitting raw data to the central server, each client performs local training using its proprietary data, sending only model updates to the server. These updates are then aggregated to refine the global model. In conventional FL frameworks, the cohort of clients engaged in training is typically static, implying the objective function is also fixed.

In practical FL systems, the dynamic nature of client arrival and departure presents significant challenges to maintaining a robust and accurate model. For instance, clients may lose interest when their local data no longer aligns with the central task, such as when a user's application shifts from text predictions to image recognition. On the other hand, clients may join when their current objectives align with those of other clients, such as when multiple users are working with similar data types or models addressing the same problem, like medical institutions collaborating on disease detection models. Additional factors, such as evolving privacy policies, or shifts in data-sharing preferences, further exacerbate these arrival and departure fluctuations, complicating the overall learning process.

**Challenges:** This dynamic FL setting introduces new challenges that are not present in traditional static FL scenarios with a fixed objective function. Specifically, the arrival and departure of clients dynamically alter the FL system's objective, as the model must adapt to the current set of clients and their associated tasks. For instance, if a client contributing unique data classes withdraws, the model no longer needs to classify those classes, fundamentally altering the training objective. Conversely,

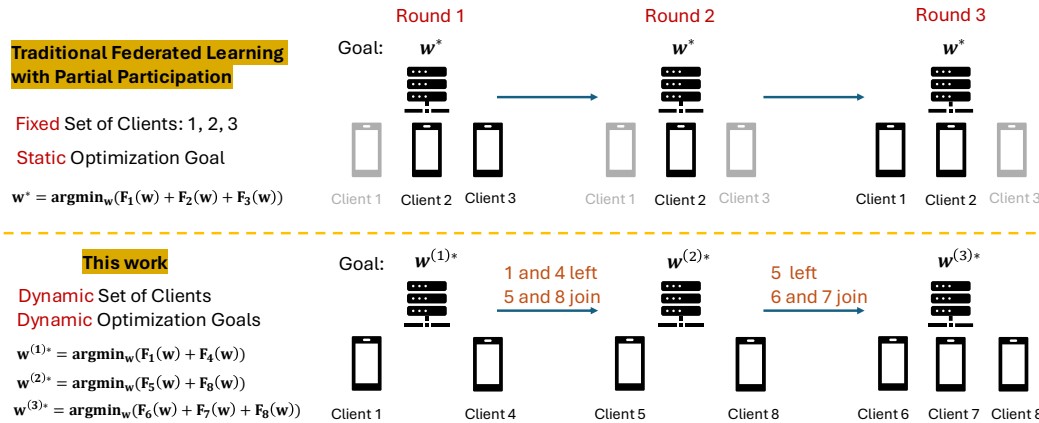

Figure 1: Comparison between traditional FL with partial participation settings and our setup. We consider both dynamic set of clients and dynamic optimization goals. In each round, our goal is to obtain a model that optimizes the loss functions of current clients. In comparison, traditional FL with partial participation approaches consider a fixed set of clients in the system with a static optimization goal as the goal is still to satisfy the clients within the current, and fixed system. Note that traditional FL with full participation is a special case of traditional FL partial participation.

the addition of clients with previously unrepresented classes necessitates model adaptation to incorporate the updated classification task. Thus, the core challenge is not merely preserving the diversity of the training set but dynamically adjusting the model to align with the evolving tasks defined by the current clients in the system. To address these shifting objectives, FL systems must be designed with the capacity for rapid and continuous adaptation, ensuring the model remains relevant, stable, and effective as the landscape of participating clients evolves.

**Illustrative Examples:** Figure 1 illustrates our system model, highlighting its key differences from traditional FL with partial client participation. In traditional FL with partial client participation, although the set of active clients may change as some clients intermittently disengage, the optimization goal remain constant throughout the training process. The objective is to converge towards the optimal model $w^*$, which minimizes the aggregated loss functions across all clients, expressed as $F_1(w) + F_2(w) + F_3(w)$. In this line of research, authors typically aim to design a client sampling strategy (e.g., (Chen et al., 2022)) or modify the aggregation weight (e.g., (Wang & Ji, 2024)) to minimize a static global loss function that remains constant over time. In contrast, our approach addresses a more intricate scenario where both the set of clients and the optimization goals evolve dynamically. In each round $g$, the set of clients changes, and the objective shifts to finding the round-specific optimal model $w^{(g)*}$, defined as the minimizer of the loss function $F_k(w)$ corresponding to the clients in that round. This dual dynamic creates substantial complexity, necessitating continuous adjustment of the optimization target to reflect the evolving composition of clients, thereby introducing challenges far beyond those encountered in traditional FL settings.

**Contributions:** Given the challenges introduced by the dynamic nature of client arrival and departure, this work makes the following key contributions to advance the field:

- We first propose a comprehensive probabilistic framework that models the formation of local datasets and classifies clients into distinct types based on their underlying probability distributions. This framework sheds light on the dynamics of how the optimal point shifts across global iterations, offering a detailed view of the impact of client variability on model performance. By examining the probabilistic relationships among client types and their associated data distributions, our approach highlights how changes in local datasets influence global optimization.

- We provide a novel theoretical analysis where an upper bound on the optimality gap is derived, quantifying the discrepancy between the global model and the theoretical optimal point. Our analysis considers a dynamic optimization goal, where each round aims to find an optimized model that minimizes the loss function of the currently active clients, in contrast to existing literature that focuses on a static goal of minimizing the loss across all clients that have ever

participated. This bound incorporates several critical factors influencing performance, including: (1) stochastic gradient noise arising from inherent randomness in local updates, (2) the number of local training iterations, which affects convergence behavior, (3) the non-IIDness of data distribution, and (4) the deviations between optimal points caused by the dynamic arrival and departure of clients. This comprehensive analysis provides a clearer understanding of the trade-offs involved in model training.

- We develop a robust algorithm for constructing an effective initial point for each training round, enabling rapid adaptation. Our approach constructs the initial model as a weighted average of previous models, with weights proportional to the similarity between the computed gradients of the models and the current set of participating clients. Leveraging these gradient-based similarities, the algorithm prioritizes models trained on clients whose data characteristics align with those of current participants, enhancing swift adaptation and mitigating performance degradation caused by dynamic client arrivals or departures. Experimental evaluations across multiple datasets demonstrate that our method achieves significant performance gains, particularly in scenarios characterized by sporadic or moderate patterns of client participation, highlighting its applicability in real-world settings.

## 2 RELATED WORK

Federated learning (FL) has emerged as a prominent paradigm for distributed machine learning, enabling the collaborative training of models across decentralized data sources while preserving data privacy (McMahan et al., 2017; Kairouz et al., 2019; Li et al., 2020). One of the foundational works introduced the Federated Averaging (FedAvg) algorithm, which remains a cornerstone of many FL systems (McMahan et al., 2017). Subsequent studies have explored various aspects of federated learning, including communication efficiency (Yang et al., 2019), robustness to adversarial attacks (Bagdasaryan et al., 2020), and personalization strategies (Smith et al., 2017). However, these approaches typically assume a fixed set of participating clients throughout the training process. Reviews of FL techniques often consider static client participation, where all clients are expected to remain available for the entire training duration (Kairouz et al., 2019; Li et al., 2020). This assumption simplifies the modeling of convergence and performance but does not adequately capture real-world scenarios characterized by dynamic client pattern. Addressing the dynamic nature of client arrival and departure is a critical gap in the current literature, motivating the need for adaptive methods that can effectively handle the entry and exit of clients during the training process.

To address the limitations of static client sets in federated learning, research on dynamic client selection and flexible participation has gained momentum, particularly in response to the challenges posed by varying client availability (Fu et al., 2023; Nishio & Yonetani, 2019; Yoshida et al., 2020; AbdulRahman et al., 2020; Martini, 2024; Li et al., 2021; Lin et al., 2021; Chai et al., 2020; Gu et al., 2021; Jhunjhunwala et al., 2022). These studies explore strategies such as optimizing client selection based on resource constraints, modeling participation patterns probabilistically, and employing adaptive algorithms to address the effects of non-IID data. The goal is to enhance overall model performance by strategically managing client participation during training, balancing computational efficiency, communication costs, and data representativeness. However, these works often assume a fixed client set, neglecting the dynamic nature of client arrival and departure. While (Ruan et al., 2021) propose a flexible federated learning framework that allows for inactive clients, incomplete updates, or dynamic client participation, their analysis is limited to scenarios where the optimization goal changes only once, specifically when a single client joins during training. This restricts the applicability of their approach to more complex and realistic patterns of client pattern.

In contrast to previous works, our approach addresses the challenge of a dynamically changing optimization goal that evolves across global rounds, focusing on the more stringent issue of dynamic client arrival and departure over time. We introduce a probabilistic framework to model the formation of local datasets, incorporating the concept of client types. These client types are characterized by the underlying probability distributions that dictate how local datasets are sampled from the global dataset. To mitigate the performance degradation caused by dynamic client arrival and departure, we propose an adaptive method for constructing the initial model. This adaptive strategy ensures robust performance, even as clients dynamically join or leave the system, leading to varying data distributions throughout training. By addressing these complexities, our framework provides a more flexible and resilient solution for federated learning in dynamic environments.

## 3 DYNAMIC FL WITH CLIENT ARRIVAL AND DEPARTURE

We consider a dynamic FL system where clients may join or leave the training process based on their interest or task needs. Clients may join when the model aligns with their objectives, when they have sufficient new data to contribute, or when the system offers financial or computational incentives. Conversely, they may leave if the global model drifts from their needs, if they lack sufficient data, or if the model's performance is not beneficial. Privacy or security concerns, such as adversarial threats or inadequate privacy guarantees, may also lead clients to leave. Additionally, clients may leave to allocate computational resources to other tasks or due to poor local model validation, while others may rejoin when they see improvements in these factors.

To mathematically characterize these scenarios, each round of FL training, denoted by $g \in \mathbb{G} = \{1, \ldots, G\}$, involves a set of clients collected in $\mathbb{K}^{(g)} = \{1, \ldots, K^{(g)}\}$. All clients are connected to a central server, with each client maintaining its own ML models tailored to specific tasks such as pattern recognition and natural language processing. These models are trained locally on client-specific data, allowing the system to leverage diverse data sources while preserving user privacy.

**Probabilistic Modeling and Definitions of Loss Functions:** To gain insights into dataset randomness and quantify data heterogeneity, we develop a probabilistic model for client types and the formation of local datasets. In this model, each local dataset $\mathbb{D}_k^{(g)}$ is considered to be sampled from a universal dataset $\mathbb{D}$. This universal dataset represents the complete set of data that could potentially be encountered throughout all training rounds. We will further quantify this probabilistic modeling in Section 4 when we present our performance analysis. The global dataset $\mathbb{D}^{(g)}$ at any training round $g$ is defined as the union of all local datasets $\mathbb{D}_k^{(g)}$ from clients $k \in \mathbb{K}^{(g)}$: $\mathbb{D}^{(g)} = \cup_{k \in \mathbb{K}^{(g)}} \mathbb{D}_k^{(g)}$. Based on this model, the local loss function of client $k$ is defined as

$$F_k^{(g)}\left(\boldsymbol{w}, \mathbb{D}_k^{(g)}\right) \triangleq \frac{\sum_{d \in \mathbb{D}} \ell(\boldsymbol{w}, d) \times \mathbf{1}\{d \in \mathbb{D}_k^{(g)}\}}{\sum_{d \in \mathbb{D}} \mathbf{1}\{d \in \mathbb{D}_k^{(g)}\}} \tag{1}$$

Here, $\mathbf{1}\{d \in \mathbb{D}_k^{(g)}\}$ is the indicator function whose value is 1 if the data point $d$ in the universal dataset belongs to the local dataset $\mathbb{D}_k^{(g)}$ and 0 otherwise. $D_k^{(g)} \triangleq \sum_{d \in \mathbb{D}} \mathbf{1}\{d \in \mathbb{D}_k^{(g)}\}$ is the size of local dataset $\mathbb{D}_k^{(g)}$, and finally $\ell(\boldsymbol{w}, d)$ measures the loss of data point $d$ in universal dataset $\mathbb{D}$ under model parameter $\boldsymbol{w}$. Similarly, let $D^{(g)} \triangleq \sum_{d \in \mathbb{D}} \mathbf{1}\{d \in \mathbb{D}^{(g)}\}$ denote the size of global dataset $\mathbb{D}^{(g)}$, we can define the global loss function as

$$F^{(g)}\left(\boldsymbol{w}, \mathbb{D}^{(g)}\right) \triangleq \frac{\sum_{d \in \mathbb{D}} \ell(\boldsymbol{w}, d) \times \mathbf{1}\{d \in \mathbb{D}^{(g)}\}}{\sum_{d \in \mathbb{D}} \mathbf{1}\{d \in \mathbb{D}^{(g)}\}} \tag{2}$$

**Local Model Training and Global Model Update:** In each round of training, the server first transmits the current global model $\boldsymbol{w}^{(g)} \in \mathbb{R}^M$ to all clients in the current round. After receiving the global model, each client $k \in \mathbb{K}^{(g)}$ updates the model with its local dataset $\mathbb{D}_k^{(g)}$ by $e_i^{(g)}$ steps of stochastic gradient descent (SGD). At SGD iteration $h \in \{0, \ldots, e_i^{(g)} - 1\}$, the update is $\boldsymbol{w}_k^{(g),h+1} = \boldsymbol{w}_k^{(g),h} - \eta^{(g)} \nabla F_k^{(g)}(\boldsymbol{w}_k^{(g),h}, \mathbb{B}_k^{(g)})$ where $\eta^{(g)}$ is the learning rate in $g$-th round, $\nabla F_k^{(g)}(\boldsymbol{w}_k^{(g),k}, \mathbb{D}_k^{(g)})$ is the gradient of client $k$'s local loss function in $g$-th round, $\mathbb{B}_k^{(g)} \subset \mathbb{D}_k^{(g)}$ is the mini-batch dataset drawn from the local dataset $\mathbb{D}_k^{(g)}$ to compute the stochastic gradient. Note that the initial point for the local model training is the current global model, i.e. $\boldsymbol{w}_k^{(g),0} = \boldsymbol{w}^{(g)}$ and we denote the final local model as $\boldsymbol{w}_k^{(g),\mathsf{F}}$. The primary objective of training an ML model is to minimize the global loss function, which directly affects the model's performance in real-time downstream tasks on clients. Dynamic client participation, where clients can join or leave the system at any time, causes the global loss functions to vary over time. Thus, the optimal global model parameters form a sequence $\{\boldsymbol{w}^{(g)*}\}_{g=1}^G$ where $\boldsymbol{w}^{(g)*} = \arg\min_{\boldsymbol{w} \in \mathbb{R}^M} F^{(g)}(\boldsymbol{w}, \mathbb{D}^{(g)}), \; g \in \mathbb{G}$. Toward this goal, after all clients finish the local model training, the sends final model $\boldsymbol{w}_k^{(g),\mathsf{F}}$ to the server, which aggregate all the final global models to update the global model in the following way: $\boldsymbol{w}^{(g+1)} = \sum_{k \in \mathbb{K}^{(g)}} \frac{D_k^{(g)} \boldsymbol{w}_k^{(g),\mathsf{F}}}{D^{(g)}}$. The server initiates the next training round by sending the updated global model $\boldsymbol{w}^{(g+1)}$ to the current set of clients $\mathbb{K}^{(g+1)}$, which includes clients that joined before

model aggregation and excludes those that left during the previous round. Unlike existing literature that aims to minimize the loss function across all potential clients, this work focuses on optimizing for the current set of participating clients, targeting $\boldsymbol{w}^{(g)*}$ at each global iteration $g$. This approach better reflects the dynamic nature of client participation, as it avoids optimizing for clients that may not rejoin the system. In our model, existing clients must complete local training before leaving, while new clients can join at any time and will participate in the next global iteration if they join after the model broadcast.

## 4 CONVERGENCE ANALYSIS

In this section, we derive a convergence bound for federated learning that accounts for dynamic client participation, including both arrivals and departures. Our analysis addresses several key factors: (1) stochastic gradient noise, which arises from the inherent randomness of local updates; (2) the impact of the number of local training iterations on convergence behavior; (3) the non-IID nature of data distribution; and (4) deviations from optimal solutions due to the dynamic nature of client participation. This analysis is grounded in widely accepted assumptions (Li et al., 2019; Ruan et al., 2021) and is framed within a probabilistic model where client types are determined by a probability distribution, which governs the random sampling process of the local dataset from the universal dataset. Our experiments include extensive tests to demonstrate that the algorithm remains robust, even when certain assumptions are not fully met.

**Definition 1 (Client Type).** *Let $\mathcal{Q}$ denote the set of probability distributions according to which global data are sampled and stored by the clients. For each client $k \in \mathbb{K}$, there exists a distribution $q \in \mathcal{Q}$ such that the probability that a local data sample $\mathrm{x}_k \in \mathbb{D}_k^{(g)}$ equals the global data sample $d \in \mathbb{D}^{(g)}$ is $p(\mathrm{x}_k = d) = q(d)$. Further, suppose there exists a set $\mathcal{S} \subset \mathbb{Z}^+$ that indexes these distributions so that $\mathcal{Q} = q_\alpha : \alpha \in \mathcal{S}$. We say client $k$ is of type $\alpha$ if the distribution of its local samples $\mathrm{x}_k \in \mathbb{D}_k^{(g)}$ is $p(\mathrm{x}_k = d) = q_\alpha(d)$.*

**Assumption 1 (Finite Device Type).** *The number of client types $S := |\mathcal{S}|$ is finite.*

**Definition 2 (Mapping from client Index to client Type).** *For each client $k \in \mathbb{K}$, we let $\tau(k) \in \{1, 2, \dots, S\}$ denote the type of client $k$.*

**Assumption 2 ($\mu$-Strong Convexity and $L$-Smoothness).** *All local loss functions $F_k^{(g)}$ and the global loss function $F^{(g)}$ are $\mu$-strongly convex and $L$-smooth (or $L$-Lipschitz continuous gradient).*

**Definition 3 (Non-IIDness Measure).** *Let $\boldsymbol{w}^{(g)*}$ be the minimizer of $F^{(g)}$ and $\boldsymbol{w}_k^{(g)*}$ be the minimizer of $F_k^{(g)}$. We can quantify the heterogeniety between the data distribution of each client and that of other clients by $\Gamma_k^{(g)} = F^{(g)}(\boldsymbol{w}_k^{(g)*}) - F_k^{(g)}(\boldsymbol{w}_k^{(g)*})$.*

**Assumption 3 (Bounded Variance of Stochastic Gradient).** *Let $\nabla F_k^{(g)}(\boldsymbol{w}, \xi)$ be the stochastic gradient at client $k$ in round $g$ given parameter $\boldsymbol{w}$ and a mini-batch $\xi$. The variance of the stochastic gradient is bounded by $\sigma_k^2$ if $\mathbb{E}_\xi \left[ \|\nabla F_k^{(g)}(\boldsymbol{w}, \xi) - \nabla F_k^{(g)}(\boldsymbol{w})\|^2 \right] \leq \sigma_k^2$.*

Before presenting the main analytical results, it is crucial to lay the groundwork with one lemma, which will provide the essential context and foundation needed to fully comprehend and derive the final outcomes. This lemma illustrates the maximum impact that variations in the set of clients—including the types of new clients that join the system and the types of clients that left—can have on shifting the new optimal point away from the previous optimal point.

**Lemma 1.** *Let $\boldsymbol{w}^{(g)*}$ be the minimizer of $F^{(g)}$ and $\boldsymbol{w}^{(g+1)*}$ be the minimizer of $F^{(g+1)}$. If for all clients $k \in \mathbb{K}^{(g)}$, all rounds $g \in \mathbb{G}$, all model parameters $\boldsymbol{w} \in \mathbb{R}^M$, and all data $d \in \mathbb{D}$, the gradient of the loss function $\nabla \ell$ is bounded on a compact set $\Omega$ which contains the possible values of the gradient during model training, i.e. $\|\nabla \ell(\boldsymbol{w}, d)\| \leq C, \ \forall \boldsymbol{w} \in \Omega$ (Reddi et al., 2021; Wang et al., 2022; Wang & Ji, 2024; Wang et al., 2024), the expected difference between two minimizers of consecutive global loss functions is bounded as follows*

$$\mathbb{E}\|\boldsymbol{w}^{(g)*} - \boldsymbol{w}^{(g+1)*}\| \leq \frac{1}{\mu}\left(\sqrt{\frac{C\pi}{12D^{(g)}}} + \sqrt{\frac{C\pi}{12D^{(g+1)}}}\right) + \frac{C}{\mu}\left|\sum_{d\in\mathbb{D}}\min\{\psi^{(g,g+1)}(d), \psi^{(g+1,g)}(d)\}\right| \quad (3)$$

*where $\psi^{(g_1,g_2)}(d)$ for any $g_1 \in \mathbb{G}$ and $g_2 \in \mathbb{G}$, $g_1 \neq g_2$ is defined as follows*

$$\psi^{(g_1,g_2)}(d) = \frac{\sum\limits_{k \in \mathbb{K}^{(g_1)}} q_{\tau(i)}(d)}{D^{(g_1)}} - \frac{\sum\limits_{i,j \in \mathbb{K}^{(g_2)}, i \neq j} q_{\tau(i)}(d)q_{\tau(j)}(d) + \sum\limits_{k \in \mathbb{K}^{(g_2)}} q_{\tau(i)}(d)}{D^{(g_2)}} \tag{4}$$

**Proof:** Please see Appendix A. $\square$ As observed in equation 3 and equation 4, the deviation of the new optimal point $\boldsymbol{w}^{(g+1)*}$ from the previous one $\boldsymbol{w}^{(g)*}$ depends solely on the conditions in rounds $g$ and $g+1$. Specifically, it is influenced by factors such as the size of the local datasets for each client, the types of clients involved, and the convexity of the global function. Notably, this deviation is independent of learning parameters such as the learning rate.

To mathematically understand the performance of our machine learning algorithm in our probabilistic framework, we need to quantify how far the model at any given training round is away from the optimal point given the client pattern dynamics. We capture this by deriving an upper bound on the optimality gap which is defined to be $\|\boldsymbol{w}^{(g)} - \boldsymbol{w}^{(g)*}\|$ in a recursive relationship.

**Theorem 1.** *If for all clients $k \in \mathbb{K}^{(g)}$, all rounds $g \in \mathbb{G}$, all model parameters $\boldsymbol{w} \in \mathbb{R}^M$, and all data $d \in \mathbb{D}$, the gradient of the loss function $\nabla \ell$ is bounded on a compact set $\Omega$ which contains the possible values of the gradient during model training, i.e. $\|\nabla \ell(\boldsymbol{w}, d)\| \leq C, \ \forall \boldsymbol{w} \in \Omega$ (Reddi et al., 2021; Wang et al., 2022; Wang & Ji, 2024; Wang et al., 2024), then we have the following recursive relationship between two consecutive optimality gaps*

$$\mathbb{E}\|\boldsymbol{w}^{(g+1)} - \boldsymbol{w}^{(g+1)*}\| \leq \underbrace{2\left(1 - \frac{1}{2}\mu\eta^{(g)}\left(\sum_{k \in \mathbb{K}^{(g)}} \frac{D_k^{(g)}e_k^{(g)}}{D^{(g)}}\right)\right)\mathbb{E}\|\boldsymbol{w}^{(g)} - \boldsymbol{w}^{(g)*}\|}_{(a)}$$

$$+ \underbrace{\left(2 + \mu\eta^{(g)}\right)C^2\left(\sum_{k \in \mathbb{K}^{(g)}} \frac{D_k^{(g)}e_k^{(g)}}{D^{(g)}}\right)\sum_{k \in \mathbb{K}^{(g)}} \frac{D_k^{(g)}\left(e_k^{(g)}-1\right)e_k^{(g)}\left(2e_k^{(g)}-1\right)}{3}}_{(b)} + \underbrace{2\eta^{(g)}\sum_{k \in \mathbb{K}^{(g)}} \frac{D_k^{(g)}\left(e_k^{(g)}\right)^2\sigma_k^2}{D^{(g)}}}_{(c)}$$

$$+ \underbrace{2\eta^{(g)}\left(\sum_{k \in \mathbb{K}^{(g)}} \frac{D_k^{(g)}\left(L\eta^{(g)}e_k^{(g)} + 2L\eta^{(g)} + 1\right)e_k^{(g)}}{D^{(g)}}\mathbb{E}[\Gamma_k^{(g)}]\right)}_{(d)} + \underbrace{\frac{2}{\mu}\left(\sqrt{\frac{C\pi}{12D^{(g)}}} + \sqrt{\frac{C\pi}{12D^{(g+1)}}}\right) + \frac{2C}{\mu}\left|\sum_{d \in \mathbb{D}}\min\{\psi^{(g,g+1)}(d), \psi^{(g+1,g)}(d)\}\right|}_{(e)}$$

$$\tag{5}$$

**Proof:** Please see Appendix B. $\square$

For any client participation pattern, regardless of how many clients join or leave the system, equation 5 captures the impact of heterogeneity on the ML performance by detailing: (1) the number of local SGD iterations $e_k^{(g)}$, (2) the non-IIDness of each client $\Gamma_k^{(g)}$, (3) local SGD noises $\sigma_k$, and (4) the size of local dataset and the types of each client in dynamic FL scenarios.

In the following, we examine each term in equation 5. Term (a) establishes the recursive relationship and explicitly identifies the factors influencing the contraction coefficient $1 - (1/2)\mu\eta^{(g)}(\sum_{k \in \mathbb{K}^{(g)}}(D_k^{(g)}e_k^{(g)}/D^{(g)}))$. A larger strong convexity coefficient $\mu$, an increased number of local SGD iterations $e_k^{(g)}$ and a higher learning rate $\eta_k^{(g)}$ all contribute to a smaller contraction coefficient. The contraction coefficient must remain between 0 and 1 to guarantee that the sequence converges. To achieve this, we need to choose $\eta^{(g)}$ and $e_k^{(g)}$ such that $0 < \min_g\{\mu\eta^{(g)}(\sum_{k \in \mathbb{K}^{(g)}} D_k^{(g)}e_k^{(g)}/D^{(g)})\} < 2$. Term (b) illustrates the influence of the number of local SGD iterations $e_k^{(g)}$ and the gradient bounding constant $C$ of the gradient. Notably, when each client performs only one local SGD (i.e. $e_k^{(g)} = 1$), the bounding constant $C$ does not affect the bound. Term (c) indicates that clients with larger SGD noise $\sigma_k$ will see a greater deviation of the model from the current optimal point when performing more local SGD iterations $e_k^{(g)}$. It is particularly concerning that the SGD noise accumulates with the square of the number of local SGD iterations, resulting in a more rapid increase in deviation. This effect is further intensified with a larger local dataset size. Term (d) highlights the impact of the non-IIDness metric $\Gamma_k^{(g)}$ and its interaction with the number of local SGD iterations. For clients with larger $\Gamma_k^{(g)}$, performing more local SGD iterations biases the local model toward the local dataset, thereby compromising the performance of the global model when aggregated. Additionally, if the function's gradient is not smooth

(i.e., if the smoothness constant $L$ is large), the non-IIDness metric will have a more pronounced effect on the optimality gap. Finally, term (e) represents the expected difference between two optimizers $\boldsymbol{w}^{(g)*}$ and $\boldsymbol{w}^{(g+1)*}$. As shown in Lemma 1, several factors can lead to a larger difference, thereby increasing the optimality gap in round $g + 1$. These factors include (1) the types of new clients joining the system, (2) the types of clients leaving, (3) the size of the global dataset in rounds $g$ and $g + 1$, and (4) the sizes of the local datasets of the joining and leaving clients.

## 5   DYNAMIC INITIAL MODEL CONSTRUCTION FOR FAST ADAPTATION

We present the key concepts of our proposed algorithm in this section, with the full pseudocode and detailed explanations available in Appendix C.

**Motivation:** Although our analysis applies to any client pattern, machine learning performance can be further improved by utilizing historical data distributions. Intuitively, if the data distribution in round $g$ closely resembles that of a previous round $g'$, where $g' < g$, initializing the global model in round $g$ with the model from round $g'$ can lead to significant performance gains. This approach leverages past knowledge to accelerate convergence and mitigates the adverse effects of sporadic or unpredictable client patterns, which often cause fluctuations in model quality. By reusing model states from rounds with similar data distributions, the learning process becomes more robust, reducing the need for the model to relearn from scratch when encountering familiar data patterns.

**Intuition:** In complex scenarios, where data distributions are heterogeneous or where no data distribution resembles the current one, it is more advantageous to initialize the model using a weighted sum of models from multiple prior rounds. The weight assigned to each model should ideally reflect the degree of similarity between data distributions in a past round $g'$ and the current round $g$. However, systematically calculating this similarity and determining the appropriate weights remains a challenging task. To address this, we propose utilizing a "pilot model" to compute gradients, which are then employed to assess similarity and derive the weights for the weighted sum.

**Initialization and Local Training:** The algorithm starts with random initialization of the global model $\boldsymbol{w}^{(0)}$. Each session is defined by a consistent data distribution and begins whenever data distribution changes due to client arrivals or departures. A session comprises at least one global round, during which the data distribution remains stable. Within each session, clients conduct local training based on the current global model, which could be a newly constructed initial model or the latest model at the server. Each client trains locally for several epochs. Upon completion, clients transmit their final local models to the server, where they are aggregated through a weighted summation. The global model from the last round of each session is preserved for future use, either for pilot model formation or constructing initial models in future sessions. The number of archived global models matches the number of completed sessions.

**Pilot Model Formation and Gradient Computation:** This step focuses on constructing the pilot model and computing gradients that accurately capture the characteristics of the current data distribution. After completing the predefined $P$ sessions in the pilot preparation stage, the pilot model $\boldsymbol{w}_p$ is constructed by averaging only those global models with good accuracy, as models with poor accuracy fail to adequately capture the underlying data distribution. Suppose there are $J$ ($\leq P$) models that perform well on the data distribution during that round, denoted by $\boldsymbol{w}^{(j)}$ for $j = 0, \ldots, J - 1$. The pilot model $\boldsymbol{w}_{\text{Pilot}}$ is then given by $\boldsymbol{w}_{\text{Pilot}} = \frac{1}{J} \sum_{j=0}^{J-1} \boldsymbol{w}^{(j)}$. It is important to note that the pilot model is computed only once throughout the algorithm's entire execution. After the pilot preparation stage, at the start of each subsequent session, additional global rounds ($V$) are conducted using $\boldsymbol{w}_{\text{Pilot}}$ as the initial global model. The difference between the updated global model $\boldsymbol{w}_{\text{Pilot}}^{(V-1)}$ and the pilot model $\boldsymbol{w}_{\text{Pilot}}$, represented as $\|\boldsymbol{w}_{\text{Pilot}}^{(V-1)} - \boldsymbol{w}_{\text{Pilot}}\|$, captures the gradients that characterize the current data distribution. These computed gradients serve as a quantitative measure of the similarity between different data distributions and are stored for future similarity assessments.

**Similarity Assessment and Dynamic Initial Model Construction:** The final step begins after at least one session has been completed following the pilot preparation stage. This step assesses the similarity between computed gradients and dynamically constructing the initial model for the current session. Similarity is evaluated by calculating the two-norm of the differences between gradients computed at the beginning of past sessions and the current gradient. Let the current gradient be

denoted as $\nabla F_{\text{Pilot}} \triangleq \|\boldsymbol{w}_{\text{Pilot}}^{(V-1)} - \boldsymbol{w}_{\text{Pilot}}\|$, and the past gradients as $\nabla F_{\text{Pilot}}^0, \ldots, \nabla F_{\text{Pilot}}^{S-1}$. A scaling factor $R$ is introduced to adjust the sensitivity of the similarity assessment, influencing the weighting of the gradient differences. These weights are used to construct the initial model for the current session as a weighted sum of previously archived models:

$$\boldsymbol{w}^{(g)} = \sum_s (w_s \times \boldsymbol{w}^{(s)}), \quad w_s = \frac{\exp\left(-R\|\nabla F_{\text{Pilot}}^s - \nabla F_{\text{Pilot}}\|_2\right)}{\sum_s \exp\left(-R\|\nabla F_{\text{Pilot}}^s - \nabla F_{\text{Pilot}}\|_2\right)} \tag{6}$$

where $w_s$ are the weights assigned to each global model $\boldsymbol{w}^{(q)}$ saved at the end of sessions after the pilot preparation stage, reflecting their similarity to the current data distribution. The `softmin` function is applied to these scaled differences, yielding normalized weights $w_s$, $s = 0, \ldots, S-1$, which sum to unity. Smaller two-norm values indicate greater similarity and result in higher weights. This approach ensures the initial model emphasizes models whose gradients closely match those of the current clients, allowing rapid adaptation and mitigating performance degradation due to the dynamic nature of client arrival and departure. By adjusting $R$, initial model construction can be controlled. When $R = 0$, the model is an average of all past saved models after the pilot preparation stage, regardless of the similarity between data distributions. In contrast, as $R \to \infty$, the initial model becomes the saved model trained on the data distribution most similar to the current one.

## 6 EXPERIMENTS

**FL Algorithm and Baseline:** We consider FedAvg (McMahan et al., 2017), FedProx (Li et al., 2020), and SCAFFOLD (Karimireddy et al., 2020) as the federated learning algorithms in our experiments to evaluate proposed algorithm. For all algorithms, the baseline continues model training using the previous model from the last round, without constructing an appropriate initial model.

**Task and Dataset:** We conducted extensive experiments to evaluate our proposed algorithm (full pseudocode provided in Algorithm 1 of Appendix C). The task of interest is image classification. We used five image datasets, ranging from the simplest to the most challenging: MNIST (LeCun et al., 1998), Fashion-MNIST (Xiao et al., 2017), SVHN (Netzer et al., 2011), CIFAR10 (Krizhevsky & Hinton, 2009), and CIFAR100 (Krizhevsky & Hinton, 2009) The models for MNIST, Fashion-MNIST, and SVHN are outlined in the Appendix D.2.

**Label Distribution:** In our experiments, we simulate a system of 10 clients. We consider four methods for distributing the labels across these clients:

- **Two-Shard**(McMahan et al., 2017; Hsu et al., 2019; Li et al., 2020; Fallah et al., 2020; Karimireddy et al., 2020)**:** For datasets with 10 labels, each label is divided into two equally-sized shards, and each client receives two different label shards. For datasets with 100 labels, the labels are divided into 10 non-overlapping batches, and each batch is split into two shards, which are then assigned to two randomly selected clients. Each client ends up with data from two labels for 10-label datasets, or 20 labels for 100-label datasets .

- **Half:** Half the clients have one half of the labels, and the rest have the other half. Each client has all the labels from their assigned half (i.e., 5 labels for 10-label datasets or 50 labels for 100-label datasets), and the data is evenly distributed, meaning that the amount of data for each label is equally divided among the clients.

- **Partial-Overlap:** Two sets of labels are selected, each containing 60% of the total labels, with a 20% overlap between them. Each client in the first set has 6 labels for 10-label datasets (or 60 labels for 100-label datasets), and each client in the second set has a similar distribution. The overlapping labels are split between the two halves of clients, with half of the data for overlapping labels going to the first set of clients and the other half to the second set. The non-overlapping labels are assigned to the clients within each set, and the data corresponding to these labels is evenly distributed across the clients, meaning that each client receives an approximately equal share of the data for their assigned labels.

- **Distinct:** Each client is assigned a unique set of labels. For datasets with 10 labels, each client receives 1 unique label, while for datasets with 100 labels, each client receives 10 unique labels.

**Test Dataset and Client Pattern:** As data distributions evolve across global rounds, the test dataset for each round consists of data with labels that represent the union of all labels held by the clients in

| FL Algorithm | Label Distribution | Dataset (Model) | 1st Transition | | 2nd Transition | | 3rd Transition | |
|---|---|---|---|---|---|---|---|---|
| | | | Proposed | Baseline | Proposed | Baseline | Proposed | Baseline |
| FedProx | Two-Shard | MNIST (MLP) | **91.71** | 87.88 | **98.45** | 96.4 | **94.77** | 90.72 |
| | | Fashion-MNIST (MLP) | **96.82** | 95.32 | **92.04** | 90.31 | **97.59** | 96.38 |
| | | SVHN (CNN) | **61.35** | 45.09 | **55.12** | 36.28 | **86.52** | 40.13 |
| | | CIFAR10 (ResNet18) | **91.77** | 83.88 | **46.26** | 35.62 | **82.58** | 58.34 |
| | | CIFAR100 (ResNet18) | **49.28** | 46.52 | **63.5** | 62.03 | **52.73** | 48.35 |
| | Half | MNIST (MLP) | **96.02** | 90.12 | **93.01** | 87.83 | **96.29** | 90.56 |
| | | Fashion-MNIST (MLP) | **86.35** | 84.6 | **90.97** | 86.02 | **86.62** | 85.06 |
| | | SVHN (CNN) | **93.37** | 10.89 | **91.45** | 20.41 | **93.51** | 61.84 |
| | | CIFAR10 (ResNet18) | **86.32** | 48.83 | **90.38** | 54.74 | **87.68** | 58.41 |
| | | CIFAR100 (ResNet18) | **62.88** | 56.16 | **64.27** | 57.14 | **63.53** | 57.41 |
| | Partial-Overlap | MNIST (MLP) | **90.92** | 86.23 | **94.09** | 89.7 | **91.32** | 87.35 |
| | | Fashion-MNIST (MLP) | **86.44** | 82.75 | **87.74** | 85.02 | **87.06** | 83.79 |
| | | SVHN (CNN) | **92.26** | 72.61 | **93.44** | 62.79 | **92.4** | 76.09 |
| | | CIFAR10 (ResNet18) | **81.5** | 77.05 | **86.99** | 83.88 | **81.69** | 79.8 |
| | | CIFAR100 (ResNet18) | **59.09** | 54.94 | **59.22** | 55.09 | **59.34** | 56.11 |
| | Distinct | MNIST (MLP) | **98.05** | 92.75 | **95.98** | 88.77 | **94.11** | 85.66 |
| | | Fashion-MNIST (MLP) | **95.5** | 92.4 | **95.8** | 90.59 | **94.02** | 87.36 |
| | | SVHN (CNN) | **95.28** | 49.68 | **92.25** | 45.18 | **87.91** | 38.45 |
| FedAvg | Two-Shard | MNIST (MLP) | **91.71** | 87.88 | **98.45** | 96.4 | **94.77** | 90.72 |
| | | Fashion-MNIST (MLP) | **96.82** | 95.32 | **92.04** | 90.31 | **97.59** | 96.38 |
| | | SVHN (CNN) | **61.35** | 45.09 | **55.12** | 36.28 | **86.52** | 40.13 |
| | | CIFAR10 (ResNet18) | **91.77** | 83.88 | **46.26** | 35.62 | **82.58** | 58.34 |
| | | CIFAR100 (ResNet18) | **49.28** | 46.52 | **63.5** | 62.03 | **52.73** | 48.35 |
| | Half | MNIST (MLP) | **96.02** | 90.12 | **93.01** | 87.83 | **96.29** | 90.56 |
| | | Fashion-MNIST (MLP) | **86.35** | 84.6 | **90.97** | 86.02 | **86.62** | 85.06 |
| | | SVHN (CNN) | **93.37** | 10.89 | **91.45** | 20.41 | **93.51** | 61.84 |
| | | CIFAR10 (ResNet18) | **86.32** | 48.83 | **90.38** | 54.74 | **87.68** | 58.41 |
| | | CIFAR100 (ResNet18) | **62.88** | 56.16 | **64.27** | 57.14 | **63.53** | 57.41 |
| | Partial-Overlap | MNIST (MLP) | **90.92** | 86.23 | **94.09** | 89.7 | **91.32** | 87.35 |
| | | Fashion-MNIST (MLP) | **86.44** | 82.75 | **87.74** | 85.02 | **87.06** | 83.79 |
| | | SVHN (CNN) | **92.26** | 72.61 | **93.44** | 62.79 | **92.4** | 76.09 |
| | | CIFAR10 (ResNet18) | **81.5** | 77.05 | **86.99** | 83.88 | **81.69** | 79.8 |
| | | CIFAR100 (ResNet18) | **59.09** | 54.94 | **59.22** | 55.09 | **59.34** | 56.11 |
| | Distinct | MNIST (MLP) | **98.05** | 92.75 | **95.98** | 88.77 | **94.11** | 85.66 |
| | | Fashion-MNIST (MLP) | **95.5** | 92.4 | **95.8** | 90.59 | **94.02** | 87.36 |
| | | SVHN (CNN) | **95.28** | 49.68 | **92.25** | 45.18 | **87.91** | 38.45 |
| SCAFFOLD | Two-Shard | MNIST (MLP) | **92.3** | 88.4 | **98.7** | 96.9 | **95.1** | 91.0 |
| | | Fashion-MNIST (MLP) | **97.1** | 95.6 | **92.5** | 91.1 | **97.9** | 96.7 |
| | | SVHN (CNN) | **62.4** | 47.0 | **56.1** | 37.5 | **87.1** | 41.3 |
| | | CIFAR10 (ResNet18) | **92.5** | 84.9 | **47.8** | 36.4 | **83.3** | 59.0 |
| | | CIFAR100 (ResNet18) | **50.3** | 47.4 | **64.0** | 62.8 | **53.4** | 49.3 |
| | Half | MNIST (MLP) | **96.5** | 91.0 | **93.5** | 88.2 | **96.8** | 91.3 |
| | | Fashion-MNIST (MLP) | **87.0** | 85.1 | **91.5** | 86.4 | **87.3** | 85.5 |
| | | SVHN (CNN) | **94.0** | 11.5 | **91.9** | 21.7 | **94.1** | 62.9 |
| | | CIFAR10 (ResNet18) | **87.0** | 49.6 | **91.1** | 55.6 | **88.3** | 59.1 |
| | | CIFAR100 (ResNet18) | **63.8** | 57.0 | **65.1** | 57.8 | **64.4** | 58.2 |
| | Partial-Overlap | MNIST (MLP) | **91.4** | 87.0 | **94.8** | 90.5 | **92.1** | 88.0 |
| | | Fashion-MNIST (MLP) | **87.2** | 83.0 | **88.5** | 85.8 | **87.6** | 84.4 |
| | | SVHN (CNN) | **93.2** | 73.8 | **94.4** | 63.8 | **93.3** | 77.1 |
| | | CIFAR10 (ResNet18) | **82.3** | 78.0 | **87.7** | 84.8 | **82.5** | 80.3 |
| | | CIFAR100 (ResNet18) | **60.0** | 55.8 | **60.2** | 55.9 | **60.5** | 57.1 |
| | Distinct | MNIST (MLP) | **98.4** | 93.5 | **96.3** | 89.5 | **94.8** | 86.3 |
| | | Fashion-MNIST (MLP) | **95.9** | 93.1 | **96.2** | 91.4 | **94.3** | 88.0 |
| | | SVHN (CNN) | **95.8** | 50.6 | **93.0** | 46.2 | **88.6** | 39.1 |

Table 1: Performance comparison of FedProx, FedAvg and SCAFFOLD under different label distributions and datasets. Performance is measured across 3 transitions for each dataset.

that round. The client patterns used in all experiments are provided in Appendix D. These patterns are designed to ensure that only a subset of the classes is present in each round.

**Results and Takeaways:** Table 1 presents a comparative analysis of the average accuracy of proposed algorithm for both FedAvg and FedProx over the first $T$ global rounds (with $T = 10$) following three shifts in data distribution. The results, encompassing all datasets, label distributions, and models, demonstrate that our algorithm effectively mitigates performance degradation caused by dynamic client arrivals and departures. Additionally, it accelerates performance recovery when the current data distribution closely resembles a previous one. Figure 2, which focuses on selected scenarios, further illustrates the advantages of our algorithm for both FedAvg and FedProx during periods of significant performance drops or boosts resulting from data distribution shifts. In both cases, the algorithm assigns higher importance to models trained on past distributions that share similarities with the current one, allowing the initial model to adapt more rapidly to the new distribution, consistently outperforming the baseline. The results confirm the effectiveness of our approach across various federated learning algorithms, datasets, models, and data distributions.

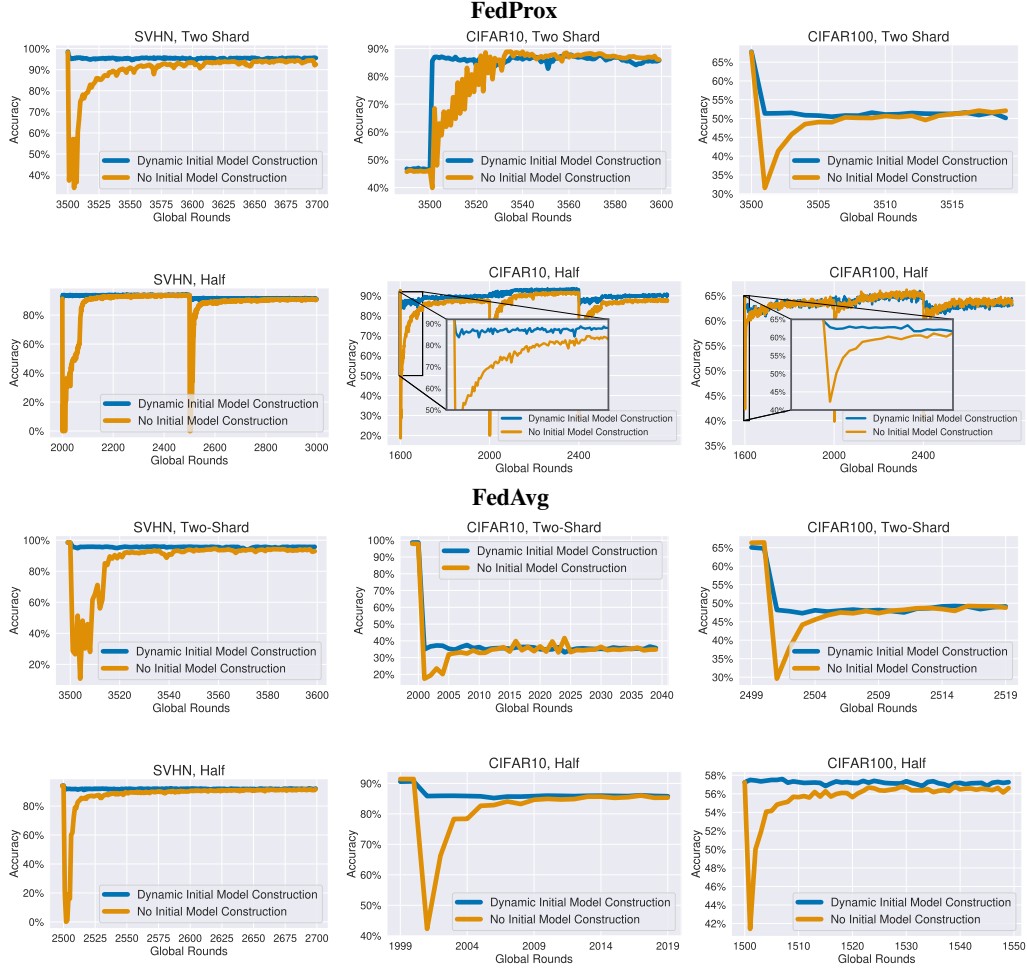

Figure 2: Performance comparison of proposed algorithm for FedProx and FedAvg with SVHN, CIFAR 10, and CIFAR 100 using Two-Shard and Half label distributions. Our proposed scheme shows robustness to the dynamic data distribution caused by dynamic client arrival and departure.

**Other Experimental Results:** Additional figures for other scenarios in Table 1 are included in Appendix D. These results demonstrate the broad applicability of proposed algorithm across different federated learning frameworks.

## 7 CONCLUSION

In this paper, we addressed the challenges of dynamic federated learning by introducing an optimization framework that adapts to dynamic client arrival and departure. Our approach accounts for these fluctuations and provides insights into how they influence shifts in optimal points. By establishing an upper bound on the optimality gap and proposing an adaptive initial model construction strategy guided by gradient similarity, we demonstrated enhanced adaptability to the current client set. Empirical results validate the robustness of our method across various datasets and dynamic client participation patterns. One promising direction for future work is to refine the initial model construction process, such that the model is only updated when beneficial or necessary, potentially reducing computational overhead while maintaining performance. This opens avenues for more efficient FL systems that can dynamically balance the trade-offs between adaptation and stability.

## 8 REPRODUCIBILITY

We utilize open-source datasets as described in Section 6. The complete mathematical proofs and details can be found in Appendix A and B. The code for training and testing is provided in the supplementary material.

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

# A    PROOF OF LEMMA 1

I first present Hoeffding's inequality which will be useful in our proof of Lemma 1.

**Fact 1** (Hoeffding's inequality). *Let $\{X_k\}_{i=1}^n$ be independent random variables such that $\mathsf{P}(X_k \in [a_k, b_k]) = 1$ for some $a_k < b_k$, and let $\epsilon > 0$, $\overline{X} = \frac{1}{n}\sum_k X_k$. Then:*

$$\mathbb{P}\left(\|\overline{X} - \mathbb{E}[\overline{X}]\| \geq \epsilon\right) \leq 2\exp\left(-\frac{12n^2\epsilon^2}{\sum_k (b_k - a_k)^2}\right) \tag{7}$$

We repeat the statement of Lemma 1 below for completeness.

**Lemma 1.** *Let $\boldsymbol{w}^{(g)*}$ be the minimizer of $F^{(g)}$ and $\boldsymbol{w}^{(g+1)*}$ be the minimizer of $F^{(g+1)}$. If for all clients $k \in \mathbb{K}^{(g)}$, all rounds $g \in \mathbb{G}$, all model parameters $\boldsymbol{w} \in \mathbb{R}^M$, and all data $d \in \mathbb{D}$, the gradient of the loss function $\nabla\ell$ is bounded on a compact set $\Omega$ which contains the possible values of the gradient during model training, i.e. $\|\nabla\ell(\boldsymbol{w}, d)\| \leq C$, $\forall \boldsymbol{w} \in \Omega$, the expected difference between two minimizers of consecutive global loss functions is bounded as follows*

$$\mathbb{E}\|\boldsymbol{w}^{(g)*} - \boldsymbol{w}^{(g+1)*}\| \leq \frac{1}{\mu}\left(\sqrt{\frac{C\pi}{12D^{(g)}}} + \sqrt{\frac{C\pi}{12D^{(g+1)}}}\right)$$
$$+ \frac{C}{\mu}\left|\sum_{d\in\mathbb{D}} \min\{\psi^{(g,g+1)}(d), \psi^{(g+1,g)}(d)\}\right| \tag{8}$$

*where $\psi^{(g_1,g_2)}(d)$ for any $g_1 \in \mathbb{G}$ and $g_2 \in \mathbb{G}$, $g_1 \neq g_2$ is defined as follows*

$$\psi^{(g_1,g_2)}(d) = \frac{\sum\limits_{k\in\mathbb{K}^{(g_1)}} q_{\tau(i)}(d)}{D^{(g_1)}} - \frac{\sum\limits_{i,j\in\mathbb{K}^{(g_2)},i\neq j} q_{\tau(i)}(d)q_{\tau(j)}(d) + \sum\limits_{k\in\mathbb{K}^{(g_2)}} q_{\tau(i)}(d)}{D^{(g_2)}} \tag{9}$$

**Proof:** By $\mu$-strong convexity, we have

$$\mu\|\boldsymbol{w}^{(g)*} - \boldsymbol{w}^{(g+1)*}\| \leq \|\nabla F^{(g+1)}(\boldsymbol{w}^{(g)*}) - \nabla F^{(g+1)}(\boldsymbol{w}^{(g+1)*})\| \tag{10}$$

Since $\nabla F^{(g+1)}(\boldsymbol{w}^{(g+1)*}) = 0 = \nabla F^{(g)}(\boldsymbol{w}^{(g)*})$, we have

$$\nabla F^{(g+1)}(\boldsymbol{w}^{(g)*}) - \nabla F^{(g+1)}(\boldsymbol{w}^{(g+1)*}) = \nabla F^{(g+1)}(\boldsymbol{w}^{(g)*}) - \nabla F^{(g)}(\boldsymbol{w}^{(g)*})$$

Combining all these, we will have

$$\mu\|\boldsymbol{w}^{(g)*} - \boldsymbol{w}^{(g+1)*}\| \leq \|\nabla F^{(g+1)}(\boldsymbol{w}^{(g)*}) - \nabla F^{(g)}(\boldsymbol{w}^{(g)*})\| \tag{11}$$

Based on equation 3, we have

$$\mathbb{E}\|\boldsymbol{w}^{(g)*} - \boldsymbol{w}^{(g+1)*}\| \leq \frac{1}{\mu}\mathbb{E}\|\nabla F^{(g+1)}(\boldsymbol{w}^{(g)*}) - \nabla F^{(g)}(\boldsymbol{w}^{(g)*})\| \tag{12}$$

Now, the goal is to derive an upper bound on $\mathbb{E}\|\nabla F^{(g+1)}(\boldsymbol{w}^{(g)*}) - \nabla F^{(g)}(\boldsymbol{w}^{(g)*})\|$. We have the following equality:

$$\|\nabla F^{(g)}(\boldsymbol{w}) - \nabla F^{(g+1)}(\boldsymbol{w})\|$$
$$= \|\nabla F^{(g)}(\boldsymbol{w}) - \mathbb{E}[\nabla F^{(g)}(\boldsymbol{w})] + \mathbb{E}[\nabla F^{(g)}(\boldsymbol{w})] - \mathbb{E}[\nabla F^{(g+1)}(\boldsymbol{w})]$$
$$+ \mathbb{E}[\nabla F^{(g+1)}(\boldsymbol{w})] - \nabla F^{(g+1)}(\boldsymbol{w})\| \tag{13}$$
$$\leq \|\nabla F^{(g)}(\boldsymbol{w}) - \mathbb{E}[\nabla F^{(g)}(\boldsymbol{w})]\| + \|\mathbb{E}[\nabla F^{(g)}(\boldsymbol{w})] - \mathbb{E}[\nabla F^{(g+1)}(\boldsymbol{w})]\|$$
$$+ \|\mathbb{E}[\nabla F^{(g+1)}(\boldsymbol{w})] - \nabla F^{(g+1)}(\boldsymbol{w})\|$$

Taking the expectation of both sides of equation 13 and using the fact that the middle term is already a scalar, we have:

$$\mathbb{E}\|\nabla F^{(g)}(\boldsymbol{w}) - \nabla F^{(g+1)}(\boldsymbol{w})\| \tag{14}$$

$$\leq \underbrace{\mathbb{E}\|\nabla F^{(g)}(\boldsymbol{w}) - \mathbb{E}[\nabla F^{(g)}(\boldsymbol{w})]\|}_{A} + \underbrace{\|\mathbb{E}[\nabla F^{(g)}(\boldsymbol{w})] - \mathbb{E}[\nabla F^{(g+1)}(\boldsymbol{w})]\|}_{B} \tag{15}$$

$$+ \underbrace{\mathbb{E}\|\nabla F^{(g+1)}(\boldsymbol{w}) - \nabla F^{(g+1)}(\boldsymbol{w})\|}_{C} \tag{16}$$

We first examine the expression for $B$. The expectation of the global loss function at round $g+1$

$$\mathbb{E}[\nabla F^{(g+1)}(\boldsymbol{w})] = \frac{1}{D^{(g+1)}} \sum_{d \in \mathbb{D}} \mathbb{E}[\nabla \ell(\boldsymbol{w}, d) \times \mathbf{1}\{d \in \mathbb{D}^{(g+1)}\}]$$
$$= \frac{1}{D^{(g+1)}} \sum_{d \in \mathbb{D}} \mathbb{E}[\nabla \ell(\boldsymbol{w}, d) \mid d \in \mathbb{D}^{(g+1)}] \times \mathsf{P}(d \in \mathbb{D}^{(g+1)}) \tag{17}$$

Similarly, the expectation of the global loss function at round $g$

$$\mathbb{E}[\nabla F^{(g)}(\boldsymbol{w})] = \frac{1}{D^{(g)}} \sum_{d \in \mathbb{D}} \mathbb{E}[\nabla \ell(\boldsymbol{w}, d) \times \mathbf{1}\{d \in \mathbb{D}^{(g)}\}]$$
$$= \frac{1}{D^{(g)}} \sum_{d \in \mathbb{D}} \mathbb{E}[\nabla \ell(\boldsymbol{w}, d) \mid d \in \mathbb{D}^{(g)}] \times \mathsf{P}(d \in \mathbb{D}^{(g)}) \tag{18}$$

The difference between the expectations is

$$\mathbb{E}[\nabla F^{(g+1)}(\boldsymbol{w})] - \mathbb{E}[\nabla F^{(g)}(\boldsymbol{w})] \tag{19}$$

$$\leq C \sum_{d \in \mathbb{D}} \left( \frac{\mathsf{P}(d \in \mathbb{D}^{(g+1)})}{D^{(g+1)}} - \frac{\mathsf{P}(d \in \mathbb{D}^{(g)})}{D^{(g)}} \right) \tag{20}$$

$$\leq C \sum_{d \in \mathbb{D}} \underbrace{\left( \frac{D^{(g)}\mathsf{P}(d \in \mathbb{D}^{(g+1)}) - D^{(g+1)}\mathsf{P}(d \in \mathbb{D}^{(g)})}{D^{(g+1)} D^{(g)}} \right)}_{B_1} \tag{21}$$

Now, we should derive an upper bound on $B_1$. We assume that there is only a total of $K$ types of clients that will appear in the system from the start to the end of training. Let $\tau(i) \in \{1, \cdots, K\}$ denote the types of clients. Based on the types of clients, we can further simplify the expressions $\mathsf{P}(d \in \mathbb{D}^{(g)})$ and $\mathsf{P}(d \in \mathbb{D}^{(g+1)})$ as follows:

$$\mathsf{P}(d \in \mathbb{D}^{(g+1)}) = \mathsf{P}(d \in \cup_k \mathbb{D}_k^{(g+1)}) \leq \sum_{k \in \mathbb{K}^{(g+1)}} \mathsf{P}(d \in \mathbb{D}_k^{(g+1)}) \tag{22}$$

$$= \sum_{k \in \mathbb{K}^{(g+1)}} \mathsf{P}(X_k = d) = \sum_{k \in \mathbb{K}^{(g+1)}} q_{\tau(i)}(d). \tag{23}$$

On the other hand, we have the following as a result of the inclusion-exclusion principle:

$$\mathsf{P}(d \in \mathbb{D}^{(g)}) = \mathsf{P}(d \in \cup_k \mathbb{D}_k^{(g)}) \tag{24}$$

$$\geq \sum_{k \in \mathbb{K}^{(g)}} \mathsf{P}(d \in \mathbb{D}_k^{(g)}) - \sum_{i,j \in \mathbb{K}^{(g)}, i \neq j} \mathsf{P}((d \in \mathbb{D}_k^{(g)}) \cap (d \in \mathbb{D}_j^{(g)})). \tag{25}$$

Since the local data samples are independently distributed, the last term of the previous inequality can be easily expressed:

$$\sum_{i,j \in \mathbb{K}^{(g)}, i \neq j} \mathsf{P}((d \in \mathbb{D}_k^{(g)}) \cap (d \in \mathbb{D}_j^{(g)})) = \sum_{i,j \in \mathbb{K}^{(g)}, i \neq j} \mathsf{P}(d \in \mathbb{D}_k^{(g)})\mathsf{P}(d \in \mathbb{D}_j^{(g)}) \tag{26}$$

$$= \sum_{i,j \in \mathbb{K}^{(g)}, i \neq j} q_{\tau(i)}(d)q_{\tau(j)}(d). \tag{27}$$

Therefore, we have:

$$\mathsf{P}(d \in \mathbb{D}^{(g)}) \geq \sum_{k \in \mathbb{K}^{(g)}} q_{\tau(i)}(d) - \sum_{i,j \in \mathbb{K}^{(g)}, i \neq j} q_{\tau(i)}(d) q_{\tau(j)}(d) \tag{28}$$

Then, we have the upper bound on $B_1$:

$$\frac{D^{(g)} \mathsf{P}(d \in \mathbb{D}^{(g+1)}) - D^{(g+1)} \mathsf{P}(d \in \mathbb{D}^{(g)})}{D^{(g+1)} D^{(g)}} \tag{29}$$

$$\leq \frac{D^{(g)} \sum_{k \in \mathbb{K}^{(g+1)}} q_{\tau(i)}(d) + D^{(g+1)} \left( \sum_{\substack{i,j \in \mathbb{K}^{(g)} \\ i \neq j}} q_{\tau(i)}(d) q_{\tau(j)}(d) - \sum_{k \in \mathbb{K}^{(g)}} q_{\tau(i)}(d) \right)}{D^{(g+1)} D^{(g)}} \tag{30}$$

$$= \psi^{(g+1,g)}(d) \tag{31}$$

Since $\|\mathbb{E}[\nabla F^{(g)}(\boldsymbol{w})] - \mathbb{E}[\nabla F^{(g+1)}(\boldsymbol{w})]\| = \|\mathbb{E}[\nabla F^{(g+1)}(\boldsymbol{w})] - \mathbb{E}[\nabla F^{(g)}(\boldsymbol{w})]\|$, we have

$$B \leq C\| \sum_{d \in \mathbb{D}} \min\{\psi^{(g+1,g)}(d), \psi^{(g,g+1)}(d)\} \| \tag{32}$$

$$\leq C\| \sum_{d \in \mathbb{D}} \min\{\psi^{(g+1,g)}(d), \psi^{(g,g+1)}(d)\} \| \tag{33}$$

Now, we derive expressions for $A$ and $C$. If we assume that gradient of the loss function is bounded by $C$ on a compact set $\Omega$, and view the Hoeffding's inequality as the complement of the Cumulative Density Function (CDF), we have the following result by using $\mathbb{E}[X] = \int_0^\infty [1 - F_X(x)]\, dx$

$$\mathbb{E}\|\nabla F^{(g)}(\boldsymbol{w}) - \mathbb{E}[\nabla F^{(g)}(\boldsymbol{w})]\| \leq \sqrt{\frac{C\pi}{12 D^{(g)}}} \tag{34}$$

Similarly,

$$\mathbb{E}\|\nabla F^{(g+1)}(\boldsymbol{w}) - \mathbb{E}[\nabla F^{(g+1)}(\boldsymbol{w})]\| \leq \sqrt{\frac{C\pi}{12 D^{(g+1)}}} \tag{35}$$

Put all things together, we have proved

$$\mathbb{E}\|\boldsymbol{w}^{(g)*} - \boldsymbol{w}^{(g+1)*}\| \leq \frac{1}{\mu} \left( \sqrt{\frac{C\pi}{12 D^{(g)}}} + \sqrt{\frac{C\pi}{12 D^{(g+1)}}} \right)$$

$$+ \frac{C}{\mu} \left| \sum_{d \in \mathbb{D}} \min\{\psi^{(g,g+1)}(d), \psi^{(g+1,g)}(d)\} \right| \tag{36}$$

$$\square$$

## B  PROOF OF THEOREM 1

We replicate the statement of Theorem 1 again for clarity.

**Theorem 1.** *If for all clients $k \in \mathbb{K}^{(g)}$, all rounds $g \in \mathbb{G}$, all model parameters $\boldsymbol{w} \in \mathbb{R}^M$, and all data $d \in \mathbb{D}$, the gradient of the loss function $\nabla \ell$ is bounded on a compact set $\Omega$ which contains the possible values of the gradient during model training, i.e. $\|\nabla \ell(\boldsymbol{w}, d)\| \leq C, \ \forall \boldsymbol{w} \in \Omega$, then we have*

*the following recursive relationship between two consecutive optimality gap*

$$\mathbb{E}\|\boldsymbol{w}^{(g+1)} - \boldsymbol{w}^{(g+1)*}\| \leq 2\left(1 - \frac{1}{2}\mu\eta^{(g)}\left(\sum_{k\in\mathbb{K}^{(g)}} \frac{D_k^{(g)}e_k^{(g)}}{D^{(g)}}\right)\right)\mathbb{E}\|\boldsymbol{w}^{(g)} - \boldsymbol{w}^{(g)*}\|$$

$$+ \left(2 + \mu\eta^{(g)}\right)C^2\left(\sum_{k\in\mathbb{K}^{(g)}} \frac{D_k^{(g)}e_k^{(g)}}{D^{(g)}}\right)\sum_{k\in\mathbb{K}^{(g)}} \frac{D_k^{(g)}}{D^{(g)}}\frac{\left(e_k^{(g)} - 1\right)e_k^{(g)}\left(2e_k^{(g)} - 1\right)}{3}$$

$$+ 2\eta^{(g)}\sum_{k\in\mathbb{K}^{(g)}} \frac{D_k^{(g)}\left(e_k^{(g)}\right)^2\sigma_k^2}{D^{(g)}} + 2\eta^{(g)}\left(\sum_{k\in\mathbb{K}^{(g)}} \frac{D_k^{(g)}\left(L\eta^{(g)}e_k^{(g)} + 2L\eta^{(g)} + 1\right)e_k^{(g)}}{D^{(g)}}\Gamma_k^{(g)}\right)$$

$$+ \frac{2}{\mu}\left(\sqrt{\frac{C\pi}{12D^{(g)}}} + \sqrt{\frac{C\pi}{12D^{(g+1)}}}\right) + \frac{2C}{\mu}\left|\sum_{d\in\mathbb{D}}\min\{\psi^{(g,g+1)}(d), \psi^{(g+1,g)}(d)\}\right| \tag{37}$$

**Proof:** By our global aggregation rules and the local model training rule, we have

$$\boldsymbol{w}^{(g+1)} = \sum_{k\in\mathbb{K}^{(g)}} \frac{D_k^{(g)}}{D^{(g)}}\left(\boldsymbol{w}^{(g)} - \eta^{(g)}\sum_{h=1}^{e_k^{(g)}}\nabla\tilde{F}_k^{(g)}\left(\boldsymbol{w}_k^{(g),h-1}\right)\right) \tag{38}$$

$$= \boldsymbol{w}^{(g)} - \eta^{(g)}\sum_{k\in\mathbb{K}^{(g)}} \frac{D_k^{(g)}}{D^{(g)}}\left(\sum_{h=1}^{e_k^{(g)}}\nabla\tilde{F}_k^{(g)}\left(\boldsymbol{w}_k^{(g),h-1}\right)\right) \tag{39}$$

$$\stackrel{\triangle}{=} \boldsymbol{w}^{(g)} - \eta^{(g)}\nabla\tilde{F}^{(g)} \tag{40}$$

where we use $\nabla\tilde{F}_k^{(g)}\left(\boldsymbol{w}_k^{(g),h-1}\right)$ to denote the stochastic gradient and $\nabla\tilde{F}^{(g)}$ to denote $\sum_{k\in\mathbb{K}^{(g)}} \frac{D_k^{(g)}}{D^{(g)}}\left(\sum_{h=1}^{e_k^{(g)}}\nabla\tilde{F}_k^{(g)}\left(\boldsymbol{w}_k^{(g),h-1}\right)\right)$ for simplicity.

Next, we relate the optimality gap at round $g + 1$ to the optimality at round $g$:

$$\|\boldsymbol{w}^{(g+1)} - \boldsymbol{w}^{(g+1)*}\|^2 \tag{41}$$

$$= \|\boldsymbol{w}^{(g+1)} - \boldsymbol{w}^{(g)*} + \boldsymbol{w}^{(g)*} - \boldsymbol{w}^{(g+1)*}\|^2 \tag{42}$$

$$\leq 2\|\boldsymbol{w}^{(g+1)} - \boldsymbol{w}^{(g)*}\|^2 + 2\|\boldsymbol{w}^{(g)*} - \boldsymbol{w}^{(g+1)*}\|^2. \tag{43}$$

Similarly, we use $\nabla F_k^{(g)}\left(\boldsymbol{w}_k^{(g),h-1}\right)$ to denote the gradient computed using the entire dataset and $\nabla F^{(g)}$ to denote $\sum_{k\in\mathbb{K}^{(g)}} \frac{D_k^{(g)}}{D^{(g)}}\left(\sum_{h=1}^{e_k^{(g)}}\nabla F_k^{(g)}\left(\boldsymbol{w}_k^{(g),h-1}\right)\right)$ for simplicity. We can expand the first term further:

$$\|\boldsymbol{w}^{(g+1)} - \boldsymbol{w}^{(g)*}\|^2 \tag{44}$$

$$= \|\boldsymbol{w}^{(g)} - \eta^{(g)}\nabla\tilde{F}^{(g)} - \boldsymbol{w}^{(g)*}\|^2 \tag{45}$$

$$= \|\boldsymbol{w}^{(g)} - \eta^{(g)}\nabla\tilde{F}^{(g)} - \boldsymbol{w}^{(g)*} - \eta^{(g)}\nabla F^{(g)} + \eta^{(g)}\nabla F^{(g)}\|^2 \tag{46}$$

$$= \|\boldsymbol{w}^{(g)} - \eta^{(g)}\nabla\tilde{F}^{(g)} - \boldsymbol{w}^{(g)*}\|^2 + (\eta^{(g)})^2\|\nabla\tilde{F}^{(g)} - \nabla F^{(g)}\|^2 \tag{47}$$

$$+ 2\eta^{(g)}\left\langle\boldsymbol{w}^{(g)} - \boldsymbol{w}^{(g)*} - \eta^{(g)}\nabla F^{(g)}, \nabla\tilde{F}^{(g)} - \nabla F^{(g)}\right\rangle \tag{48}$$

$$= \|\boldsymbol{w}^{(g)} - \boldsymbol{w}^{(g)*}\|^2 - 2\eta^{(g)}\left\langle\boldsymbol{w}^{(g)} - \boldsymbol{w}^{(g)*}, \nabla F^{(g)}\right\rangle + (\eta^{(g)})^2\|\nabla F^{(g)}\|^2 \tag{49}$$

$$+ (\eta^{(g)})^2\|\nabla\tilde{F}^{(g)} - \nabla F^{(g)}\|^2 + 2\eta^{(g)}\left\langle\boldsymbol{w}^{(g)} - \boldsymbol{w}^{(g)*} - \eta^{(g)}\nabla F^{(g)}, \nabla\tilde{F}^{(g)} - \nabla F^{(g)}\right\rangle \tag{50}$$

$$= \|\boldsymbol{w}^{(g)} - \boldsymbol{w}^{(g)*}\|^2 \underbrace{-2\eta^{(g)}\left\langle\boldsymbol{w}^{(g)} - \boldsymbol{w}^{(g)*}, \nabla F^{(g)}\right\rangle}_{A_1} \underbrace{+(\eta^{(g)})^2\|\nabla F^{(g)}\|^2}_{A_2} \tag{51}$$

$$\underbrace{+(\eta^{(g)})^2\|\nabla\tilde{F}^{(g)}-\nabla F^{(g)}\|^2}_{A_3}\underbrace{+2\eta^{(g)}\left\langle\boldsymbol{w}^{(g)}-\boldsymbol{w}^{(g)*}-\eta^{(g)}\nabla F^{(g)},\nabla\tilde{F}^{(g)}-\nabla F^{(g)}\right\rangle}_{A_4}. \quad (52)$$

Note that $\mathbb{E}[A_4]=0$ because $\mathbb{E}[\nabla\tilde{F}^{(g)}-\nabla F^{(g)}]=0$. Now, let's expand $A_1$ and $A_2$:

$$A_2=(\eta^{(g)})^2\|\sum_{k\in\mathbb{K}^{(g)}}\frac{D_k^{(g)}}{D^{(g)}}\sum_{h=1}^{e_k^{(g)}}\nabla F_k^{(g)}(\boldsymbol{w}_k^{(g),h-1})\|^2 \quad (53)$$

$$\leq(\eta^{(g)})^2\sum_{k\in\mathbb{K}^{(g)}}\frac{D_k^{(g)}}{D^{(g)}}\|\sum_{h=1}^{e_k^{(g)}}\nabla F_k^{(g)}(\boldsymbol{w}_k^{(g),h-1})\|^2 \quad (54)$$

$$\leq(\eta^{(g)})^2\sum_{k\in\mathbb{K}^{(g)}}\frac{D_k^{(g)}}{D^{(g)}}e_k^{(g)}\sum_{h=1}^{e_k^{(g)}}\|\nabla F_k^{(g)}(\boldsymbol{w}_k^{(g),h-1})\|^2 \quad (55)$$

$$\leq2L(\eta^{(g)})^2\sum_{k\in\mathbb{K}^{(g)}}\frac{D_k^{(g)}e_k^{(g)}}{D^{(g)}}\sum_{h=1}^{e_k^{(g)}}(\nabla F_k^{(g)}(\boldsymbol{w}_k^{(g),h-1})-\nabla F_k^{(g)*}). \quad (56)$$

Now, expanding $A_1$:

$$A_1=-2\eta^{(g)}\langle\boldsymbol{w}^{(g)}-\boldsymbol{w}^{(g)*},\nabla F^{(g)}\rangle \quad (57)$$

$$=-2\eta^{(g)}\sum_{k\in\mathbb{K}^{(g)}}\frac{D_k^{(g)}}{D^{(g)}}\sum_{h=1}^{e_k^{(g)}}\langle\boldsymbol{w}^{(g)}-\boldsymbol{w}^{(g)*},\nabla F_k^{(g)}(\boldsymbol{w}_k^{(g),h-1})\rangle \quad (58)$$

$$=\underbrace{-2\eta^{(g)}\sum_{k\in\mathbb{K}^{(g)}}\frac{D_k^{(g)}}{D^{(g)}}\sum_{h=1}^{e_k^{(g)}}\langle\boldsymbol{w}^{(g)}-\boldsymbol{w}_k^{(g),h-1},\nabla F_k^{(g)}(\boldsymbol{w}_k^{(g),h-1})\rangle}_{A_{11}} \quad (59)$$

$$\underbrace{-2\eta^{(g)}\sum_{k\in\mathbb{K}^{(g)}}\frac{D_k^{(g)}}{D^{(g)}}\sum_{h=1}^{e_k^{(g)}}\langle\boldsymbol{w}_k^{(g),h-1}-\boldsymbol{w}^{(g)*},\nabla F_k^{(g)}(\boldsymbol{w}_k^{(g),h-1})\rangle}_{A_{12}}. \quad (60)$$

We then expand $A_{11}$ and $A_{12}$:

$$A_{11}=-2\eta^{(g)}\sum_{k\in\mathbb{K}^{(g)}}\frac{D_k^{(g)}}{D^{(g)}}\sum_{h=1}^{e_k^{(g)}}\left\langle\boldsymbol{w}_k^{(g)}-\boldsymbol{w}_k^{(g),h-1},\nabla F_k^{(g)}(\boldsymbol{w}_k^{(g),h-1})\right\rangle \quad (61)$$

$$\leq\sum_{k\in\mathbb{K}^{(g)}}\frac{D_k^{(g)}}{D^{(g)}}\sum_{h=1}^{e_k^{(g)}}2\eta^{(g)}\left\|\boldsymbol{w}_k^{(g)}-\boldsymbol{w}_k^{(g),h-1}\right\|\left\|\nabla F_k^{(g)}(\boldsymbol{w}_k^{(g),h-1})\right\| \quad (62)$$

$$\leq\sum_{k\in\mathbb{K}^{(g)}}\frac{D_k^{(g)}}{D^{(g)}}\sum_{h=1}^{e_k^{(g)}}\eta^{(g)}\left(\frac{1}{\eta^{(g)}}\left\|\boldsymbol{w}_k^{(g)}-\boldsymbol{w}_k^{(g),h-1}\right\|^2+\eta^{(g)}\left\|\nabla F_k^{(g)}(\boldsymbol{w}_k^{(g),h-1})\right\|^2\right) \quad (63)$$

$$\leq\sum_{k\in\mathbb{K}^{(g)}}\frac{D_k^{(g)}}{D^{(g)}}\sum_{h=1}^{e_k^{(g)}}\left(\left\|\boldsymbol{w}_k^{(g)}-\boldsymbol{w}_k^{(g),h-1}\right\|^2+(\eta^{(g)})^2\left\|\nabla F_k^{(g)}(\boldsymbol{w}_k^{(g),h-1})\right\|^2\right). \quad (64)$$

$$A_{12} = -2\eta^{(g)} \sum_{k \in \mathbb{K}^{(g)}} \frac{D_k^{(g)}}{D^{(g)}} \sum_{h=1}^{e_k^{(g)}} \left\langle \boldsymbol{w}_k^{(g),h-1} - \boldsymbol{w}^{(g)*}, \nabla F_k^{(g)}(\boldsymbol{w}_k^{(g),h-1}) \right\rangle \tag{65}$$

$$\leq -2\eta^{(g)} \sum_{k \in \mathbb{K}^{(g)}} \frac{D_k^{(g)}}{D^{(g)}} \sum_{h=1}^{e_k^{(g)}} \left( F_k^{(g)}(\boldsymbol{w}_k^{(g),h-1}) - F_k^{(g)}(\boldsymbol{w}_k^{(g)*}) + \frac{\mu}{2} \left\| \boldsymbol{w}_k^{(g),h-1} - \boldsymbol{w}^{(g)*} \right\|^2 \right). \tag{66}$$

Combining $A_{11}$ and $A_{12}$, we have:

$$A_1 = \sum_{k \in \mathbb{K}^{(g)}} \frac{D_k^{(g)}}{D^{(g)}} \sum_{h=1}^{e_k^{(g)}} \left( \|\boldsymbol{w}^{(g)} - \boldsymbol{w}_k^{(g),h-1}\|^2 + (\eta^{(g)})^2 \|\nabla F_k^{(g)}(\boldsymbol{w}_k^{(g),h-1})\|^2 \right. \tag{67}$$

$$\left. -2\eta^{(g)} \left( F_k^{(g)}(\boldsymbol{w}_k^{(g),h-1}) - F_k^{(g)}(\boldsymbol{w}^{(g)*}) - \eta^{(g)}\mu\|\boldsymbol{w}_k^{(g),h-1} - \boldsymbol{w}^{(g)*}\|^2 \right) \right). \tag{68}$$

Next, we combine $A_1$ and $A_2$:

$$A_1 + A_2 = \underbrace{2L(\eta^{(g)})^2 \sum_{k \in \mathbb{K}^{(g)}} \frac{D_k^{(g)}}{D^{(g)}}(e_k^{(g)}+1) \sum_{h=1}^{e_k^{(g)}} \left( F_k^{(g)}(\boldsymbol{w}_k^{(g)}) - F_k^{(g)*} \right)}_{B_1 \text{ (first term)}} \tag{69}$$

$$\underbrace{-2\eta^{(g)} \sum_{k \in \mathbb{K}^{(g)}} \frac{D_k^{(g)}}{D^{(g)}} \left( \sum_{h=1}^{e_k^{(g)}} \left( F_k^{(g)}(\boldsymbol{w}_k^{(g),h-1}) - F_k^{(g)}(\boldsymbol{w}^{(g)*}) \right) \right)}_{B_1 \text{ (second term)}} \tag{70}$$

$$+ \sum_{k \in \mathbb{K}^{(g)}} \frac{D_k^{(g)}}{D^{(g)}} \sum_{h=1}^{e_k^{(g)}} \|\boldsymbol{w}^{(g)} - \boldsymbol{w}_k^{(g),h-1}\|^2 \tag{71}$$

$$\underbrace{-\eta^{(g)}\mu \sum_{k \in \mathbb{K}^{(g)}} \frac{D_k^{(g)}}{D^{(g)}} \sum_{h=1}^{e_k^{(g)}} \|\boldsymbol{w}_k^{(g),h-1} - \boldsymbol{w}^{(g)*}\|^2}_{B_2}. \tag{72}$$

For the third item, we further simplify:

$$\|\boldsymbol{w}_k^{(g),h-1} - \boldsymbol{w}^{(g)*}\|^2 = \|\boldsymbol{w}_k^{(g),h-1} - \boldsymbol{w}^{(g)} + \boldsymbol{w}^{(g)} - \boldsymbol{w}^{(g)*}\|^2 \tag{73}$$

$$= \|\boldsymbol{w}_k^{(g),h-1} - \boldsymbol{w}^{(g)}\|^2 + \|\boldsymbol{w}^{(g)} - \boldsymbol{w}^{(g)*}\|^2 \tag{74}$$

$$+ 2\langle \boldsymbol{w}_k^{(g),h-1} - \boldsymbol{w}^{(g)}, \boldsymbol{w}^{(g)} - \boldsymbol{w}^{(g)*} \rangle \tag{75}$$

$$\geq \|\boldsymbol{w}_k^{(g),h-1} - \boldsymbol{w}^{(g)}\|^2 + \|\boldsymbol{w}^{(g)} - \boldsymbol{w}^{(g)*}\|^2 \tag{76}$$

$$- 2\|\boldsymbol{w}_k^{(g),h-1} - \boldsymbol{w}^{(g)}\| \cdot \|\boldsymbol{w}^{(g)} - \boldsymbol{w}^{(g)*}\| \tag{77}$$

$$\geq \|\boldsymbol{w}_k^{(g),h-1} - \boldsymbol{w}^{(g)}\|^2 + \|\boldsymbol{w}^{(g)} - \boldsymbol{w}^{(g)*}\|^2 \tag{78}$$

$$- 2\|\boldsymbol{w}_k^{(g),h-1} - \boldsymbol{w}^{(g)}\| - \frac{1}{2}\|\boldsymbol{w}^{(g)} - \boldsymbol{w}^{(g)*}\|^2 \tag{79}$$

$$\geq -\|\boldsymbol{w}_k^{(g),h-1} - \boldsymbol{w}^{(g)}\| + \frac{1}{2}\|\boldsymbol{w}^{(g)} - \boldsymbol{w}^{(g)*}\|^2. \tag{80}$$

From this, we derive an upper bound on $B_2$:

$$B_2 \leq -\frac{\eta^{(g)}\mu}{2} \left( \sum_{k \in \mathbb{K}^{(g)}} \frac{D_k^{(g)} e_k^{(g)}}{D^{(g)}} \right) \|\boldsymbol{w}^{(g)} - \boldsymbol{w}^{(g)*}\|^2 \tag{81}$$

$$+ \mu\eta^{(g)} \sum_{k \in \mathbb{K}^{(g)}} \frac{D_k^{(g)}}{D^{(g)}} \sum_{h=1}^{e_k^{(g)}} \|\boldsymbol{w}_k^{(g),h-1} - \boldsymbol{w}^{(g)}\|^2. \tag{82}$$

Plug the expressions for the third item and $B_2$ back into $A_1 + A_2$:

$$A_1 + A_2 \leq B_1 + (1 + \mu\eta^{(g)}) \sum_{k \in \mathbb{K}^{(g)}} \frac{D_k^{(g)}}{D^{(g)}} \sum_{h=1}^{e_k^{(g)}} \|\boldsymbol{w}_k^{(g),h-1} - \boldsymbol{w}^{(g)}\|^2 \tag{83}$$

$$- \frac{\eta^{(g)}\mu}{2} \left( \sum_{k \in \mathbb{K}^{(g)}} \frac{D_k^{(g)} e_k^{(g)}}{D^{(g)}} \right) \|\boldsymbol{w}^{(g)} - \boldsymbol{w}^{(g)*}\|^2. \tag{84}$$

We then expand $B_1$ as follows:

$$B_1 = 2L(\eta^{(g)})^2 \sum_{k \in \mathbb{K}^{(g)}} \frac{D_k^{(g)}}{D^{(g)}} (e_k^{(g)} + 1) \sum_{h=1}^{e_k^{(g)}} \left( F_k^{(g)}(\boldsymbol{w}_k^{(g)}) - F_k^{(g)*} \right) \tag{85}$$

$$+ \sum_{k \in \mathbb{K}^{(g)}} \frac{D_k^{(g)}}{D^{(g)}} \sum_{h=1}^{e_k^{(g)}} \underbrace{\underbrace{\left( 2L(\eta^{(g)})^2(e_k^{(g)} + 1) - 2\eta^{(g)} \right)}_{-V_k^{(g)}} \left( F_k^{(g)}(\boldsymbol{w}_k^{(g),h-1}) - F_k^{(g)}(\boldsymbol{w}^{(g)*}) \right)}_{B_{11}}. \tag{86}$$

We expand $B_{11}$ further:

$$B_{11} = -V_k^{(g)} \left( F_k^{(g)}(\boldsymbol{w}_k^{(g),h-1}) - F_k^{(g)}(\boldsymbol{w}^{(g)*}) \right) \tag{87}$$

$$= -V_k^{(g)} \left( F_k^{(g)}(\boldsymbol{w}_k^{(g),h-1}) - F_k^{(g)}(\boldsymbol{w}^{(g)}) + F_k^{(g)}(\boldsymbol{w}^{(g)}) - F_k^{(g)}(\boldsymbol{w}^{(g)*}) \right) \tag{88}$$

$$\leq -V_k^{(g)} \left( \left\langle \nabla F_k^{(g)}(\boldsymbol{w}^{(g)}), \boldsymbol{w}_k^{(g),h-1} - \boldsymbol{w}^{(g)} \right\rangle + \frac{\mu}{2} \|\boldsymbol{w}_k^{(g),h-1} - \boldsymbol{w}^{(g)}\|^2 \right) \tag{89}$$

$$- V_k^{(g)} \left( F_k^{(g)}(\boldsymbol{w}^{(g)}) - F_k^{(g)}(\boldsymbol{w}^{(g)*}) \right) \tag{90}$$

$$\leq V_k^{(g)} \left\langle \boldsymbol{w}^{(g)} - \boldsymbol{w}_k^{(g),h-1}, \nabla F_k^{(g)}(\boldsymbol{w}^{(g)}) \right\rangle - \frac{\mu V_k^{(g)}}{2} \|\boldsymbol{w}_k^{(g),h-1} - \boldsymbol{w}^{(g)}\|^2 \tag{91}$$

$$- V_k^{(g)} \left( F_k^{(g)}(\boldsymbol{w}^{(g)}) - F_k^{(g)}(\boldsymbol{w}^{(g)*}) \right) \tag{92}$$

$$\leq \frac{V_k^{(g)}\eta^{(g)}}{2} \|\nabla F_k^{(g)}(\boldsymbol{w}^{(g)})\|^2 + \frac{V_k^{(g)}}{2\eta^{(g)}} \|\boldsymbol{w}^{(g)} - \boldsymbol{w}_k^{(g),h-1}\|^2 \tag{93}$$

$$- \frac{\mu V_k^{(g)}}{2} \|\boldsymbol{w}_k^{(g),h-1} - \boldsymbol{w}^{(g)}\|^2 - V_k^{(g)} \left( F_k^{(g)}(\boldsymbol{w}^{(g)}) - F_k^{(g)}(\boldsymbol{w}^{(g)*}) \right) \tag{94}$$

$$\leq V_k^{(g)} L\eta^{(g)} \left( F_k^{(g)}(\boldsymbol{w}^{(g)}) - F_k^{(g)*} \right) + \frac{V_k^{(g)}(1 - \mu\eta^{(g)})}{2\eta^{(g)}} \|\boldsymbol{w}^{(g)} - \boldsymbol{w}_k^{(g),h-1}\|^2 \tag{95}$$

$$- V_k^{(g)} \left( F_k^{(g)}(\boldsymbol{w}^{(g)}) - F_k^{(g)}(\boldsymbol{w}^{(g)*}) \right) \tag{96}$$

$$\leq V_k^{(g)} L \eta^{(g)} \left( F_k^{(g)}(\boldsymbol{w}^{(g)*}) - F_k^{(g)*} \right) + \|\boldsymbol{w}^{(g)} - \boldsymbol{w}_k^{(g),h-1}\|^2 \tag{97}$$

$$+ V_k^{(g)}(1 - L\eta^{(g)}) \left( F_k^{(g)}(\boldsymbol{w}^{(g)*}) - F_k^{(g)}(\boldsymbol{w}^{(g)}) \right) \tag{98}$$

$$\leq V_k^{(g)} L \eta^{(g)} \left( F_k^{(g)}(\boldsymbol{w}^{(g)*}) - F_k^{(g)*} \right) + \|\boldsymbol{w}^{(g)} - \boldsymbol{w}_k^{(g),h-1}\|^2 \tag{99}$$

$$+ V_k^{(g)}(1 - L\eta^{(g)}) \left( F_k^{(g)}(\boldsymbol{w}^{(g)*}) - F_k^{(g)*} \underbrace{+ F_k^{(g)*} - F_k^{(g)}(\boldsymbol{w}^{(g)})}_{\leq 0} \right) \tag{100}$$

$$\leq V_k^{(g)} L \eta^{(g)} \left( F_k^{(g)}(\boldsymbol{w}^{(g)*}) - F_k^{(g)*} \right) + \|\boldsymbol{w}^{(g)} - \boldsymbol{w}_k^{(g),h-1}\|^2 \tag{101}$$

$$+ V_k^{(g)} \left( F_k^{(g)}(\boldsymbol{w}^{(g)*}) - F_k^{(g)*} \right) \tag{102}$$

$$\leq V_k^{(g)} \left( L\eta^{(g)} + 1 \right) \left( F_k^{(g)}(\boldsymbol{w}^{(g)*}) - F_k^{(g)*} \right) + \|\boldsymbol{w}^{(g)} - \boldsymbol{w}_k^{(g),h-1}\|^2. \tag{103}$$

Substituting the expression for $B_{11}$ back into $B_1$ and noting that $V_k^{(g)} \leq 2\eta^{(g)}$, we have:

$$B_1 = 2L(\eta^{(g)})^2 \sum_{k \in \mathbb{K}^{(g)}} \frac{D_k^{(g)}}{D^{(g)}} (e_k^{(g)} + 1) \sum_{h=1}^{e_k^{(g)}} \left( F_k^{(g)}(\boldsymbol{w}^{(g)*}) - F_k^{(g)*} \right) \tag{104}$$

$$+ \sum_{k \in \mathbb{K}^{(g)}} \frac{D_k^{(g)}}{D^{(g)}} \sum_{h=1}^{e_k^{(g)}} 2\eta^{(g)} \left( L\eta^{(g)} + 1 \right) \left( F_k^{(g)}(\boldsymbol{w}^{(g)*}) - F_k^{(g)*} \right) \tag{105}$$

$$+ \sum_{k \in \mathbb{K}^{(g)}} \frac{D_k^{(g)}}{D^{(g)}} \sum_{h=1}^{e_k^{(g)}} \|\boldsymbol{w}^{(g)} - \boldsymbol{w}_k^{(g),h-1}\|^2 \tag{106}$$

$$= 2\eta^{(g)} \sum_{k \in \mathbb{K}^{(g)}} \frac{D_k^{(g)} \left( L\eta^{(g)} e_k^{(g)} + 2L\eta^{(g)} + 1 \right)}{D^{(g)}} \sum_{h=1}^{e_k^{(g)}} \left( F_k^{(g)}(\boldsymbol{w}^{(g)*}) - F_k^{(g)*} \right) \tag{107}$$

$$+ \sum_{k \in \mathbb{K}^{(g)}} \frac{D_k^{(g)}}{D^{(g)}} \sum_{h=1}^{e_k^{(g)}} \|\boldsymbol{w}^{(g)} - \boldsymbol{w}_k^{(g),h-1}\|^2 \tag{108}$$

$$= 2\eta^{(g)} \sum_{k \in \mathbb{K}^{(g)}} \frac{D_k^{(g)} \left( L\eta^{(g)} e_k^{(g)} + 2L\eta^{(g)} + 1 \right)}{D^{(g)}} e_k^{(g)} \Gamma_k^{(g)} \tag{109}$$

$$+ \sum_{k \in \mathbb{K}^{(g)}} \frac{D_k^{(g)}}{D^{(g)}} \sum_{h=1}^{e_k^{(g)}} \|\boldsymbol{w}^{(g)} - \boldsymbol{w}_k^{(g),h-1}\|^2. \tag{110}$$

Substituting the expression back into $A_1 + A_2$:

$$A_1 + A_2 \leq 2\eta^{(g)} \sum_{k \in \mathbb{K}^{(g)}} \frac{D_k^{(g)} \left( L\eta^{(g)} e_k^{(g)} + 2L\eta^{(g)} + 1 \right)}{D^{(g)}} e_k^{(g)} \Gamma_k^{(g)} \tag{111}$$

$$+ (2 + \mu\eta^{(g)}) \sum_{k \in \mathbb{K}^{(g)}} \frac{D_k^{(g)}}{D^{(g)}} \sum_{h=1}^{e_k^{(g)}} \|\boldsymbol{w}^{(g)} - \boldsymbol{w}_k^{(g),h-1}\|^2 \tag{112}$$

$$- \frac{\eta^{(g)}\mu}{2} \left( \sum_{k \in \mathbb{K}^{(g)}} \frac{D_k^{(g)} e_k^{(g)}}{D^{(g)}} \right) \|\boldsymbol{w}^{(g)} - \boldsymbol{w}^{(g)*}\|^2. \tag{113}$$

Next, we bound the term $\sum_{h=1}^{e_k^{(g)}} \|\boldsymbol{w}^{(g)} - \boldsymbol{w}_k^{(g),h-1}\|^2$:

$$\sum_{h=1}^{e_k^{(g)}} \|\boldsymbol{w}^{(g)} - \boldsymbol{w}_k^{(g),h-1}\|^2 = \sum_{h=2}^{e_k^{(g)}} \| \sum_{m=0}^{h-2} \nabla F_k^{(g)}(\boldsymbol{w}_k^{(g),m})\|^2 \tag{114}$$

$$= \sum_{q=0}^{e_k^{(g)}-2} \| \sum_{m=0}^{q} \nabla F_k^{(g)}(\boldsymbol{w}_k^{(g),m})\|^2 \leq \sum_{q=0}^{e_k^{(g)}-2} (q+1) \sum_{m=0}^{q} \|\nabla F_k^{(g)}(\boldsymbol{w}_k^{(g),m})\|^2 \tag{115}$$

$$= \sum_{q=0}^{e_k^{(g)}-2} (q+1)^2 C^2 = C^2 \sum_{q=0}^{e_k^{(g)}-2} (q+1)^2 = \frac{C^2(e_k^{(g)}-1)e_k^{(g)}(2e_k^{(g)}-1)}{6}. \tag{116}$$

Substituting into $A_1 + A_2$, we get:

$$A_1 + A_2 \leq 2\eta^{(g)} \sum_{k \in \mathbb{K}^{(g)}} \frac{D_k^{(g)} \left( L\eta^{(g)}e_k^{(g)} + 2L\eta^{(g)} + 1 \right)}{D^{(g)}} e_k^{(g)} \Gamma_k^{(g)} \tag{117}$$

$$+ (2 + \mu\eta^{(g)}) \sum_{k \in \mathbb{K}^{(g)}} \frac{D_k^{(g)}}{D^{(g)}} \frac{C^2(e_k^{(g)}-1)e_k^{(g)}(2e_k^{(g)}-1)}{6} \tag{118}$$

$$- \frac{\mu\eta^{(g)}}{2} \left( \sum_{k \in \mathbb{K}^{(g)}} \frac{D_k^{(g)} e_k^{(g)}}{D^{(g)}} \right) \|\boldsymbol{w}^{(g)} - \boldsymbol{w}^{(g)*}\|^2. \tag{119}$$

Finally, let's derive the expression for $A_3$:

$$A_3 = (\eta^{(g)})^2 \left\| \sum_{k \in \mathbb{K}^{(g)}} \frac{D_k^{(g)}}{D^{(g)}} \sum_{h=1}^{e_k^{(g)}} \left( \nabla \tilde{F}_k^{(g)} - \nabla F_k^{(g)} \right) \right\|^2 \tag{120}$$

$$\leq (\eta^{(g)})^2 \sum_{k \in \mathbb{K}^{(g)}} \frac{D_k^{(g)}}{D^{(g)}} \left\| \sum_{h=1}^{e_k^{(g)}} \left( \nabla \tilde{F}_k^{(g)} - \nabla F_k^{(g)} \right) \right\|^2 \tag{121}$$

$$\leq \eta^{(g)} \sum_{k \in \mathbb{K}^{(g)}} \frac{D_k^{(g)}}{D^{(g)}} e_k^{(g)} \sum_{h=1}^{e_k^{(g)}} \left\| \nabla \tilde{F}_k^{(g)} - \nabla F_k^{(g)} \right\|^2 \tag{122}$$

$$\leq \eta^{(g)} \sum_{k \in \mathbb{K}^{(g)}} \frac{D_k^{(g)}}{D^{(g)}} (e_k^{(g)})^2 \sigma_k^2. \tag{123}$$

Taking the expectation and combining the expressions for $A_1$, $A_2$, and $A_3$, we have:

$$\mathbb{E}\|\boldsymbol{w}^{(g+1)} - \boldsymbol{w}^{(g+1)*}\| \leq 2 \left( 1 - \frac{1}{2}\mu\eta^{(g)} \left( \sum_{k \in \mathbb{K}^{(g)}} \frac{D_k^{(g)} e_k^{(g)}}{D^{(g)}} \right) \right) \mathbb{E}\|\boldsymbol{w}^{(g)} - \boldsymbol{w}^{(g)*}\| \tag{124}$$

$$+ \left(2 + \mu\eta^{(g)}\right) C^2 \left(\sum_{k \in \mathbb{K}^{(g)}} \frac{D_k^{(g)} e_k^{(g)}}{D^{(g)}}\right) \sum_{k \in \mathbb{K}^{(g)}} \frac{D_k^{(g)}}{D^{(g)}} \frac{\left(e_k^{(g)} - 1\right) e_k^{(g)} \left(2e_k^{(g)} - 1\right)}{3} \tag{125}$$

$$+ 2\eta^{(g)} \sum_{k \in \mathbb{K}^{(g)}} \frac{D_k^{(g)} \left(e_k^{(g)}\right)^2 \sigma_k^2}{D^{(g)}} + 2\eta^{(g)} \left(\sum_{k \in \mathbb{K}^{(g)}} \frac{D_k^{(g)} \left(L\eta^{(g)} e_k^{(g)} + 2L\eta^{(g)} + 1\right) e_k^{(g)}}{D^{(g)}} \Gamma_k^{(g)}\right) \tag{126}$$

$$+ 2\mathbb{E}\|\boldsymbol{w}^{(g)*} - \boldsymbol{w}^{(g+1)*}\|. \tag{127}$$

Plugging the expression in Lemma 1 for the last terms yields

$$\mathbb{E}\|\boldsymbol{w}^{(g+1)} - \boldsymbol{w}^{(g+1)*}\| \leq 2 \left(1 - \frac{1}{2}\mu\eta^{(g)} \left(\sum_{k \in \mathbb{K}^{(g)}} \frac{D_k^{(g)} e_k^{(g)}}{D^{(g)}}\right)\right) \mathbb{E}\|\boldsymbol{w}^{(g)} - \boldsymbol{w}^{(g)*}\| \tag{128}$$

$$+ \left(2 + \mu\eta^{(g)}\right) C^2 \left(\sum_{k \in \mathbb{K}^{(g)}} \frac{D_k^{(g)} e_k^{(g)}}{D^{(g)}}\right) \sum_{k \in \mathbb{K}^{(g)}} \frac{D_k^{(g)}}{D^{(g)}} \frac{\left(e_k^{(g)} - 1\right) e_k^{(g)} \left(2e_k^{(g)} - 1\right)}{3} \tag{129}$$

$$+ 2\eta^{(g)} \sum_{k \in \mathbb{K}^{(g)}} \frac{D_k^{(g)} \left(e_k^{(g)}\right)^2 \sigma_k^2}{D^{(g)}} + 2\eta^{(g)} \left(\sum_{k \in \mathbb{K}^{(g)}} \frac{D_k^{(g)} \left(L\eta^{(g)} e_k^{(g)} + 2L\eta^{(g)} + 1\right) e_k^{(g)}}{D^{(g)}} \Gamma_k^{(g)}\right) \tag{130}$$

$$+ \frac{2}{\mu} \left(\sqrt{\frac{C\pi}{12 D^{(g)}}} + \sqrt{\frac{C\pi}{12 D^{(g+1)}}}\right) + \frac{2C}{\mu} \left|\sum_{d \in \mathbb{D}} \min\{\psi^{(g,g+1)}(d), \psi^{(g+1,g)}(d)\}\right| \tag{131}$$

$\square$

## C  Dynamic Initial Model Construction for Fast Adaptation

**Algorithm Details**: Algorithm 1 outlines the pseudocode for our proposed "dynamic initial model construction for fast adaptation" algorithm. In this context, a "session" differs from the "number of global rounds." A new round is initiated whenever there is a change in data distribution, such as clients joining or leaving, while the set of clients remains constant within a round. Each round comprises at least one global iteration. The implementation of Algorithm 1 requires specifying the number of global iterations per round $T$, the number of rounds for pilot model preparation $P$, the number of rounds dedicated to model training $S$, and the number of global iterations used to compute the gradient reflecting the characteristics of the current dataset $V$.

Initially, in line 1, training begins with a randomly initialized global weight $\boldsymbol{w}^{(0)}$, and several lists are initialized: $Q_1$ to store trained models, $Q_2$ to store computed gradients, and $Q_3$ to store the two-norm values of differences between gradients. From lines 23 to 29, each client in the current set performs local model training based on the current global model, which may be either a new initial model or the latest model at the server. Upon completion of local training, each client transmits its final local model to the server, where global aggregation is performed using a weighted sum. From lines 30 to 32, the final global models are saved at the end of each round, either for pilot model computation or as components of a new initial model.

Lines 33 to 35 describe the formation of the pilot model. When the pilot preparation stage concludes—i.e., when the length of $Q_1$ reaches $P$—the average of all models in the current $Q_1$ is taken to form the pilot model, $\boldsymbol{w}_p$. In lines 3 to 21, following the pilot model preparation stage (for $g = 0, \ldots, PT - 1$), an additional $V$ global iterations are conducted at the start of each round using the pilot model $\boldsymbol{w}_p$ to compute the difference between the final global model, $\boldsymbol{w}_p^{(V-1)}$, and the pilot model $\boldsymbol{w}_p$. This difference, $\boldsymbol{w}_p^{(V-1)} - \boldsymbol{w}_p$, reflects a combination of gradients computed from mini-batches of local datasets across all current clients. These gradients encapsulate the data characteristics, making them representative of the datasets and useful for evaluating the similarity between different client sets.

---

**Algorithm 1** Dynamic Initial Model Construction for Fast Adaptation

---

**Input:** The number of global rounds within one round $T$, the number of sessions in pilot model preparation stage $P$, the number of sessions to execute $S(\geq P)$, the number of global rounds $V$ to compute the gradient used to calculate similarity.

1: **Initialize:** Randomly initialize global model $\boldsymbol{w}^{(0)}$, a list $Q_1$ to store trained global models, and a list $Q_2$ to store gradients that will be used to calculate similarity, a list $Q_3$ to store the two-norm values between gradients, a similarity scaling factor $R$.

2: **for** $g = 0, \ldots, ST - 1$ **do**

3:     **if** $g \geq PT$ and $g\%T == 0$ **then**

4:         **for** $g' = 0, \ldots, V - 1$ **do**

5:             **if** $g' == 0$ **then**

6:                 $\boldsymbol{w}_p^{(g')} \leftarrow \boldsymbol{w}_p$

7:             **end if**

8:             **for** $k \in \mathbb{K}^{(g)}$ **do**

9:                 **for** $h \in \{0, \ldots, e_k^{(g)} - 1\}$ **do**

10:                     Perform local model training based on $\boldsymbol{w}_p^{(g')}$.

11:                 **end for**

12:                 Send the final local model $\boldsymbol{w}_{k,p}^{(g'),\mathsf{F}}$ to the server.

13:             **end for**

14:             The servers perform the aggregation using weighted summation to get $\boldsymbol{w}_p^{(g'+1)}$.

15:         **end for**

16:         **if** $|Q_2| > 0$ **then**

17:             Compute the two-norm values of the difference between $\boldsymbol{w}_p^{(V-1)} - \boldsymbol{w}_p$ and every element in $Q_2$. Multiply all results by $R$ and store them to $Q_3$.

18:             $Q_4 \leftarrow Softmin(Q_3)$

19:             $\boldsymbol{w}^{(g)} \leftarrow Q_4[0] \times \boldsymbol{w}^{((P+1)T-1)} + \cdots + Q_4[(g - (P+1)T)//T - 1] \times \boldsymbol{w}^{(g-1)}$

20:         **end if**

21:         Append $\boldsymbol{w}_p^{(V-1)} - \boldsymbol{w}_p$ to $Q_2$.

22:     **end if**

23:     **for** $k \in \mathbb{K}^{(g)}$ **do**

24:         **for** $h \in \{0, \ldots, e_k^{(g)} - 1\}$ **do**

25:             Perform local model training using SGD based on $\boldsymbol{w}^{(g)}$.

26:         **end for**

27:         Send the final local model $\boldsymbol{w}_k^{(g),\mathsf{F}}$ to the server.

28:     **end for**

29:     The servers perform the aggregation using weighted summation to get $\boldsymbol{w}^{(g+1)}$.

30:     **if** $g\%T == T - 1$ **then**

31:         Append $\boldsymbol{w}^{(g)}$ to $Q_1$

32:     **end if**

33:     **if** $|Q_1| == P$ **then**

34:         The pilot model $\boldsymbol{w}_p \leftarrow$ The average of all models in $Q_1$.

35:     **end if**

36: **end for**

---

Lines 16 to 19 focus on computing both the similarity and the new initial model. Specifically, in line 17, the two-norm of the difference between two gradients is used to represent similarity; smaller two-norm values indicate greater similarity between client sets. We introduce a constant $R$ to control the emphasis on differences among the two-norm values. The function of $R$ becomes evident in lines 18 and 19. In line 18, the `softmin` function is applied to normalize the values and generate a set of weights that sum to 1, assigning higher weights to smaller two-norm values. These weights are stored in $Q_4$. In line 19, the new initial model is computed as a weighted sum of all models from the pilot preparation stage, using weights from $Q_4$. The role of $R$ is critical here: a high value of $R$ results in one dominant weight after the `softmin` operation, favoring the model trained on the set of clients most similar to the current one. Conversely, a lower value of $R$ leads to a more balanced distribution of weights, reflecting the differences in two-norm values. Finally, in line 21, the computed gradient is saved in $Q_2$ for future similarity assessments.

| FL Algorithm | Label Distribution | Dataset (Model) | 1st Transition | | 2nd Transition | | 3rd Transition | |
|---|---|---|---|---|---|---|---|---|
| | | | Proposed | Baseline | Proposed | Baseline | Proposed | Baseline |
| **FedProx** | **Half** | TinyImageNet (ResNet34) | **80.71** | 72.8 | **77.93** | 72.28 | **80.31** | 73.36 |
| **FedAvg** | **Half** | TinyImageNet (ResNet34) | **80.72** | 72.81 | **77.77** | 72.24 | **80.43** | 73.42 |
| | **Partial-Overlap** | TinyImageNet (ResNet34) | **92.27** | 79.11 | **93.67** | 82.01 | **92.48** | 77.2 |

Table 2: Performance comparison of FedProx, FedAvg under different label distributions for Tiny ImageNet dataset. Performance is measured across 3 transitions for each dataset.

## D   MORE EXPERIMENT RESULTS AND DETAILS

Table 2 is the average accuracy for the first 10 accuracy following three shifts in data distributions. Figure 3 presents more results for FedProx with various label distributions, datasets and models.

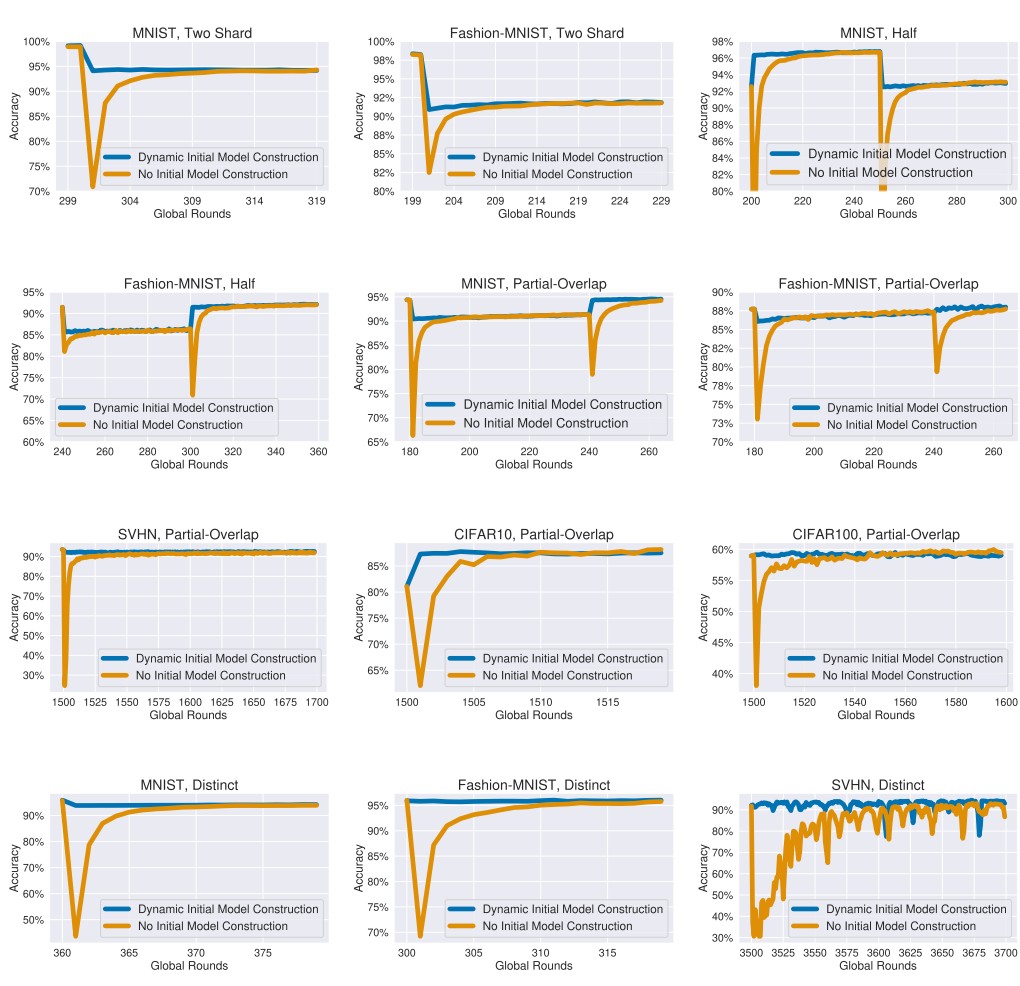

Figure 3: Performance comparison of proposed algorithm for FedProx to the baseline across the remaining examined label distributions, datasets, and models.

Figure 4 presents more results for FedAvg with various label distributions, datasets and models.

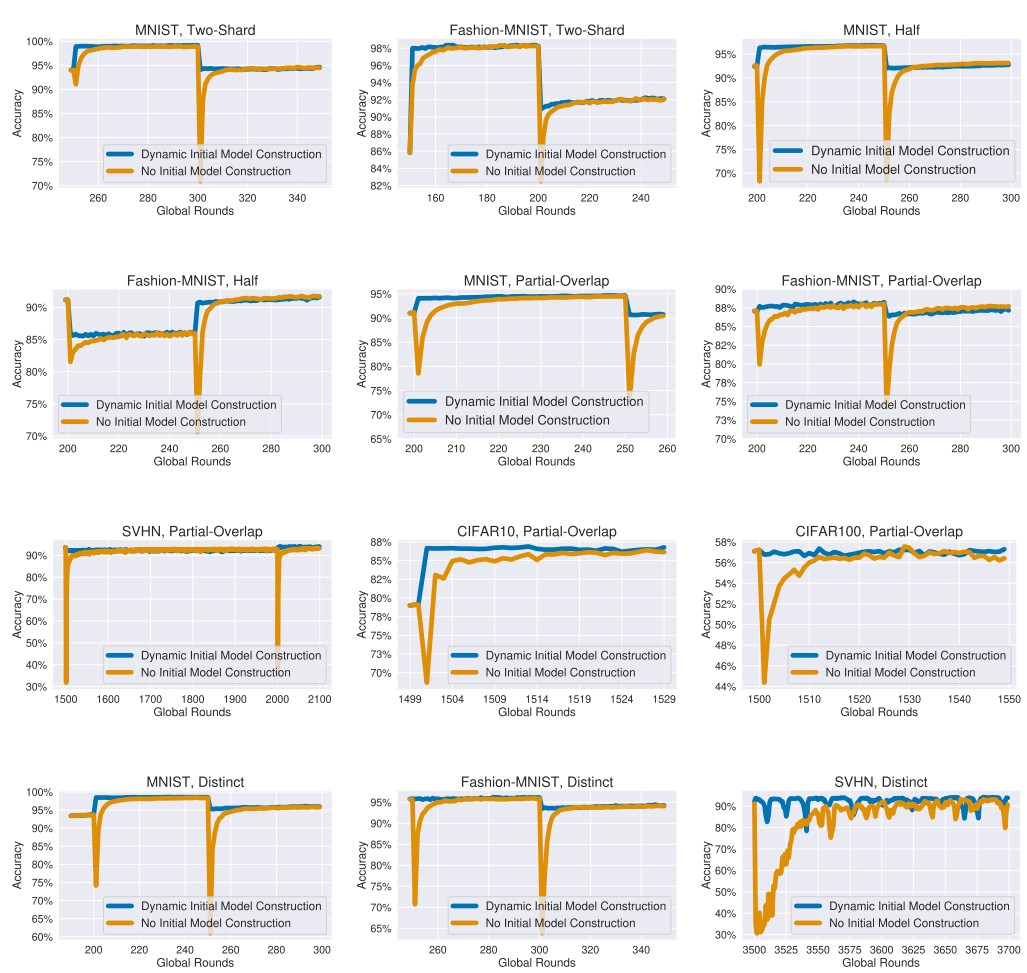

Figure 4: Performance comparison of proposed algorithm for FedProx to the baseline across the remaining examined label distributions, datasets, and models.

## D.1 CLIENT PATTERN

For client pattern used in all label distributions, please see Table 3 and 4. Client pattern is the same for FedAvg and FedProx.

| Session | Two Shard | | | | | Distinct | | |
|---|---|---|---|---|---|---|---|---|
| | MNIST | Fashion-MNIST | SVHN | CIFAR10 | CIFAR100 | MNIST | Fashion-MNIST | SVHN |
| 1 | [0, 4, 6, 7] | [0, 1, 4, 9] | [1, 5, 6] | [1, 5, 6] | [1, 5, 6] | [0, 1, 2] | [0, 1, 2] | [0, 1, 2] |
| 2 | [5] | [5] | [0, 4, 8] | [0, 4, 8] | [0, 2, 3, 9] | [0, 1, 2] | [0, 1, 2] | [0, 1, 2] |
| 3 | [0, 4, 6] | [0, 4, 6, 8] | [3] | [1, 2, 3, 5, 6, 9] | [8] | [3, 4, 5] | [3, 4, 5] | [3, 4, 5] |
| 4 | [5] | [5] | [1, 4] | [4] | [0, 2, 3, 4, 6, 9] | [6, 7, 8, 9] | [6, 7, 8, 9] | [6, 7, 8, 9] |
| 5 | [4, 6, 7] | [0, 4, 6, 7] | [3, 5, 8] | [1, 2, 3, 5, 6, 8] | [5, 8] | [0, 1, 2] | [0, 1, 2] | [0, 1, 2] |
| 6 | [5] | [5] | [1, 4, 7] | [4] | [0, 2, 3, 6, 7] | [3, 4, 5] | [3, 4, 5] | [3, 4, 5] |
| 7 | [4, 6] | [4, 6] | [5] | [1, 5, 6, 9] | [1, 5, 8] | [6, 7, 8, 9] | [6, 7, 8, 9] | [6, 7, 8, 9] |
| 8 | [5] | [5] | [1, 4] | [0, 4] | [0, 2, 7] | [3, 4, 5] | [3, 4, 5] | [3, 4, 5] |

Table 3: Client Pattern for Label Distribution Two-Shard and Distinct

| Session | Half & Partial-Overlap | | | | |
|---|---|---|---|---|---|
| | MNIST | Fashion-MNIST | SVHN | CIFAR10 | CIFAR100 |
| 1 | [0, 1, 2, 3, 4] | [0, 1, 2, 3, 4] | [0, 1, 2, 3, 4] | [0, 1, 2, 3, 4] | [0, 1, 2, 3, 4] |
| 2 | [5, 6, 7, 8, 9] | [5, 6, 7, 8, 9] | [5, 6, 7, 8, 9] | [5, 6, 7, 8, 9] | [5, 6, 7, 8, 9] |
| 3 | [0, 1, 2, 3, 4] | [0, 1, 2, 3, 4] | [0, 1, 2, 3, 4] | [0, 1, 2, 3, 4] | [0, 1, 2, 3, 4] |
| 4 | [5, 6, 7, 8, 9] | [5, 6, 7, 8, 9] | [5, 6, 7, 8, 9] | [5, 6, 7, 8, 9] | [5, 6, 7, 8, 9] |
| 5 | [0, 1, 2, 3, 4] | [0, 1, 2, 3, 4] | [0, 1, 2, 3, 4] | [0, 1, 2, 3, 4] | [0, 1, 2, 3, 4] |
| 6 | [5, 6, 7, 8, 9] | [5, 6, 7, 8, 9] | [5, 6, 7, 8, 9] | [5, 6, 7, 8, 9] | [5, 6, 7, 8, 9] |
| 7 | [0, 1, 2, 3, 4] | [0, 1, 2, 3, 4] | [0, 1, 2, 3, 4] | [0, 1, 2, 3, 4] | [0, 1, 2, 3, 4] |
| 8 | [5, 6, 7, 8, 9] | [5, 6, 7, 8, 9] | [5, 6, 7, 8, 9] | [5, 6, 7, 8, 9] | [5, 6, 7, 8, 9] |

Table 4: Client Pattern for Label Distribution Half and Partial-Overlap

## D.2 MODELS FOR MNIST, FASHION-MNIST, AND SVHN

The model code for MNIST, Fashion-MNIST, and SVHN is as follows.

```python
class net(nn.Module):
    def __init__(self, dataset_name) -> None:
        super().__init__()
        if dataset_name == "mnist":
            self.in_channel = 28 * 28
        elif dataset_name == "fmnist":
            self.in_channel = 28 * 28
        self.out_channel = 10
        self.net = nn.Linear(self.in_channel, self.out_channel)

    def forward(self, x):
        x = x.view(-1, x.shape[1] * x.shape[2] * x.shape[3])
        x = self.net(x)
        return nn.functional.log_softmax(x, dim=1)
```

Listing 1: Model for MNIST and Fashion-MNIST

```python
class CNN_SVHN(nn.Module):
    def __init__(self, num_classes=10):
        super().__init__()
        self.conv1 = nn.Conv2d(in_channels=3, out_channels=32,
            kernel_size=3, padding=1)
        self.conv2 = nn.Conv2d(in_channels=32, out_channels=64,
            kernel_size=3, padding=1)
        self.conv3 = nn.Conv2d(in_channels=64, out_channels=128,
            kernel_size=3, padding=1)
        self.fc1 = nn.Linear(128 * 4 * 4, 256)
        self.fc2 = nn.Linear(256, num_classes)
```

```
 9          self.dropout = nn.Dropout(0.5)  # Dropout with a probability
                of 0.5
10
11      def forward(self, x):
12          x = F.relu(F.max_pool2d(self.conv1(x), 2))
13          x = F.relu(F.max_pool2d(self.conv2(x), 2))
14          x = F.relu(F.max_pool2d(self.conv3(x), 2))
15          x = x.view(x.size(0), -1)
16          x = F.relu(self.fc1(x))
17          x = self.dropout(x)  # Apply Dropout after the first fully
                connected layer
18          x = self.fc2(x)
19          return x
```

Listing 2: Model for SVHN

