# OpenReview forum: "Federated Learning with Dynamic Client Arrival and Departure: Convergence and Rapid Adaptation via Initial Model Construction"
_ICLR.cc/2025/Conference — Submitted to ICLR 2025_

### Official Review · Reviewer_iAPF · 2024-10-21

**Soundness:** 2
**Presentation:** 3
**Contribution:** 2
**Rating:** 3
**Confidence:** 5

**Summary:**

This work considers FL with dynamic participation where clients may leave or join the system in any global iteration and analyzes the model convergence in this setup. This work further proposes an adaptive initial model construction strategy that employs weighted averaging guided by gradient similarity, which is validated on various datasets.

**Strengths:**

(1) The considered dynamic setup is less explored in existing works but is quite crucial in practical applications.

(2) The theorem and analysis in section 4 are comprehensive.

(3) The proposed adaptive algorithm shows benefits compared with baselines.

**Weaknesses:**

(1) The theoretical results are under strong assumptions such as strong convexity (in Assumption 2), bounded gradient (in Theorem 1).

(2) Theorem 1 only captures a resursive relationship between two consecutive optimality gap. How does this theorem show the convergence of this dynamic system or in which condition may the FL algorithm diverge?

(3) With the initial model in (6), how does the convergence change from Section 4?

(4) The authors mentioned (Ruan et al. 2021a) in related works, but there is no comparison between this work and proposed method. Also, (Ruan et al. 2021a) and (Ruan et al. 2021b) are the same reference.

(5) The challenges in Line 50 mentioned that some classes may not be met in some iterations; can the client participation in Table 2 and Table 3 guarantee this discription?

(6) The setups "half" and "partial-overlap" seem to be similar with those used in client participation works. Given this, it is more convincing to compare the proposed algorithm with those methods, such as (Gu et al.， 2021) (Jhunjhunwala et al. 2022).

**Questions:**

Besides of the above weakneess, could authors further explain the effect of scaling factor R in Line 380? What if we don't use this scaling factor?

---

> ### Author Response · Authors · 2024-11-21
> **Weakness 1**
>
> Thank you for your valuable question. In highly dynamic scenarios, where the optimization objective changes from round to round, the strong convexity assumption can be crucial for gaining analytical insights. For instance, prior work such as [4] also relies on the strong convexity assumption, even in a simplified setting where only a single new device is allowed to join the system throughout the training process. This setting is less complex than ours, which involves continuous client arrivals and departures. That said, we acknowledge the value of exploring extensions to non-convex settings, and we plan to investigate this direction in future work.
>
> Regarding the bounded gradient assumption, it is often essential for dynamic systems that employ adaptive methodologies, as noted in prior studies [2,3,5,6]. In our specific context, this assumption is necessary to establish a quantitative relationship between the data distribution and the deviation between two optimal global models. To the best of our knowledge, our work is among the first to explicitly connect data distribution characteristics with model deviation in a federated learning setting. Nevertheless, we are committed to exploring how our theoretical analysis could be extended to relax these assumptions, further enhancing its applicability to a broader range of scenarios. We have cited the aforementioned work using bounded gradient assumptions in our Lemma 1 and Theorem 1 statements on Pages 5 and 6.
>
> [2] Wang, Shiqiang, and Mingyue Ji. "A Lightweight Method for Tackling Unknown Participation Statistics in Federated Averaging." The Twelfth International Conference on Learning Representations, 2024.
>
> [3] Wang, Yujia, et al. "FADAS: Towards Federated Adaptive Asynchronous Optimization." arXiv preprint arXiv:2407.18365 (2024).
>
> [4] Ruan, Yichen, et al. "Towards flexible device participation in federated learning." International Conference on Artificial Intelligence and Statistics. PMLR, 2021.
>
> [5] Reddi, Sashank J., et al. "Adaptive Federated Optimization." International Conference on Learning Representations, 2021.
>
> [6] Wang, Yujia, Lu Lin, and Jinghui Chen. "Communication-efficient adaptive federated learning." International Conference on Machine Learning. PMLR, 2022.

---

> ### Author Response · Authors · 2024-11-21
> **Weakness 2**
>
> Based on our theorem, we can expand the recursive relationship up to the initial optimality gap $\mathbb{E}\| \mathbf{w}^{(0)} - \mathbf{w}^{(0)\ast}\|$. The left-hand side of the inequality would be $\mathbb{E}\| \mathbf{w}^{(g)} - \mathbf{w}^{(g)\ast}\|$. The first term on the right-hand side is $r_0 r_1 \cdots r_{g-1} \times \mathbb{E}\| \mathbf{w}^{(0)} - \mathbf{w}^{(0)\ast}\|$, where $r_m = \left(1 - \frac{1}{2}\mu\eta^{(m)}\left(\sum_{k \in \mathbb{K}^{(m)}} \frac{D_k^{(m)} e_k^{(m)}}{D^{(m)}}\right)\right)$ for $m = \{0, \dots, g-1\}$. All other terms on the right-hand side follow the structure $a_0 + r_1 a_1 + r_1 r_2 a_2 + \dots + r_1 r_2 \cdots r_{g-1} a_{g-1}$. For example, for term (c) on the right-hand side, we have $a_m = 2\eta^{(m)} \sum_{k \in \mathbb{K}^{(m)}} \frac{D_k^{(m)} \left(e_k^{(m)}\right)^2 \sigma_k^2}{D^{(m)}}$. To guarantee convergence, we need to ensure that $0 < r_m < 1$. That is, for all $m = 0, \cdots,  g -1$, we require  $0 < \mu \eta^{(m)} \left( \sum_{k \in \mathbb{K}^{(m)}} \frac{D_k^{(m)} e_k^{(m)}}{D^{(m)}} \right) < 2.$
> If this condition is violated, the algorithm will diverge.

---

> ### Author Response · Authors · 2024-11-21
> **Weakness 3**
>
> By constructing the initial model, we anticipate faster convergence empirically. This is because the constructed model is primarily composed of the global model that has already learned a data distribution similar to the current one. In Theorem 1, term (e) represents the deviation between two optimal global models. When the data distributions are similar, these optimal models should not deviate significantly from each other. Additionally, the global model that has already learned the previous data distribution will be close to the optimal model for the current round. As a result, constructing the initial model helps to initialize the training process in a way that leads to quicker convergence.

---

> ### Author Response · Authors · 2024-11-21
> **Weakness 4**
>
> Thank you for pointing out the issue with the references. We have corrected it. The key differences between our work and [4] are as follows:
>
> First and most importantly, our analytical results apply to any client arrival and departure patterns, whereas their main result is limited to a simpler scenario where only one new client joins the system at a time throughout the training process. This fundamental distinction allows our approach to handle more complex and dynamic client participation, making it more versatile and applicable to real-world federated learning scenarios.
>
> Second, their proposed fast reboot algorithm is designed to be effective when only one new client joins at a time. While it is possible that their algorithm could be extended to more complex scenarios, the performance in such cases remains unclear and is not explicitly addressed.
>
> Finally, we provide a quantitative analysis of how data distributions impact the deviation between two optimal global models, as shown in Lemma 1. In contrast, they quantify this impact using an indirect non-IIDness measure in their Theorem 3.2. Our approach offers a more direct and precise analysis of the influence of data distribution shifts on model performance.
>
> [4] Ruan, Yichen, et al. "Towards flexible device participation in federated learning." International Conference on Artificial Intelligence and Statistics. PMLR, 2021.

---

> ### Author Response · Authors · 2024-11-21
> **Weakness 5**
>
> Thank you for your question, which provides us an opportunity to clarify this point. In all of our experimental results, each iteration involves only a subset of the available classes. For example, in the “half” label distribution scenario, each iteration includes just 50\% of the classes from the dataset. Specifically, this corresponds to 50 classes for the CIFAR-100 dataset, and 5 classes for the MNIST, Fashion-MNIST, SVHN, and CIFAR-10 datasets. In the “Two Shard” label distribution scenario (as shown in session 2 of Table 2), the client set for that session consists solely of client 5, which holds data from only two classes for 10-class dataset and only 20 classes for 100-class dataset. This approach ensures that, in every iteration, only a portion of the classes is present, addressing the challenges related to class availability mentioned in Line 50. In our updated paper, we note on Page 9 that these patterns are designed to ensure that only a subset of classes is included in each round.

---

> ### Author Response · Authors · 2024-11-21
> **Weakness 6**
>
> Thank you for highlighting these two works. While the “half” and “partial-overlap” setups in our study may appear similar to those used in previous client participation works, the system model we consider differs substantially from those in the referenced papers. We have elaborated on these differences in Figure 1 of our manuscript.
>
> In both of the cited works, the global loss function remains fixed throughout the training process, which is clearly stated in their problem setups (e.g., Equation 1 in both papers) and reflected in their analytical results. For example, in Theorem 5.1 of [7], the global loss function  $f$  does not change across rounds, and similarly, in Theorem 2 of [8], the global loss function remains constant.
>
> In contrast, our framework accounts for the dynamic nature of the system, where clients can join or leave at any time, causing the optimization goal to evolve from one round to the next. This dynamic setting introduces new challenges and requires a different approach, which is not directly comparable to the static setup in the referenced works.
>
> [7] Gu, Xinran, et al. "Fast federated learning in the presence of arbitrary device unavailability." Advances in Neural Information Processing Systems 34 (2021): 12052-12064.
>
> [8] Jhunjhunwala, Divyansh, et al. "Fedvarp: Tackling the variance due to partial client participation in federated learning." Uncertainty in Artificial Intelligence. PMLR, 2022.

---

> > ### Comment · Reviewer_iAPF · 2024-11-25
> > **Thank you for responses**
> >
> > Thank authors for your responses.
> >
> > For weakness 2, I suggest you add the final convergence (with the initial optimal gap in RHS) to the manuscript and discuss the convergence condition based on this theorem.
> >
> > For weakness 5, providing the references of the simulated label distribution may help the readers to understand easily.
> >
> > For Weakness 6, I still think the current baseline is not sufficient.
> >
> > My main concern about this work is the strong assumption and weak comparisons in experiments. After reading others' reviews, I would keep the current score.

---

> ### Author Response · Authors · 2024-11-21
> **Question**
>
> Thank you for your question. The scaling factor $R$ is introduced to control the contribution of previous global models when constructing the initial model. When $R = 0$, the initial model is simply the average of all previously saved models, regardless of the similarity between their data distributions. On the other hand, as $R \to \infty$, the initial model increasingly prioritizes the global model trained on the data distribution most similar to the current one, giving higher weight to that model. This means that a larger $R$ effectively assigns more importance to the model that has been trained on a distribution similar to the current one.
>
> The scaling factor $R$ can be adjusted to accommodate different datasets, as varying levels of sensitivity to data distribution may be needed depending on the characteristics of the dataset. If we omit the scaling factor, we would not scale the difference in the similarity between data distributions. In that case, the original values of the similarity measure (the negative two-norm of the gradient differences) would determine the weights assigned to the global models when constructing the initial model. We have highlighted the sentences on Page 8 discussing the effect of $R$.

---

### Official Review · Reviewer_jhbc · 2024-10-31

**Soundness:** 3
**Presentation:** 2
**Contribution:** 3
**Rating:** 5
**Confidence:** 4

**Summary:**

This paper presents a FL framework that addresses the challenge of dynamic client participation, a scenario in which clients frequently join or exit the training process. A probabilistic model is developed to quantify the impact of client arrival and departure on the FL optimization objective, providing an upper bound on the optimality gap under these dynamic conditions. Furthermore, a gradient-based approach is introduced for constructing an adaptive initial model that aligns more closely with the data distributions of the active clients.

**Strengths:**

+ This work addresses a key limitation in existing FL research by considering dynamic client participation, a more realistic scenario than the often-assumed static client set. The proposed framework, which adjusts for client variability, adds flexibility and may lead to increased model adaptability.

+ The theoretical derivation of an optimality gap bound, considering factors like stochastic gradient noise and non-IID data distributions, is a valuable contribution. This analysis provides a comprehensive understanding of how dynamic client changes impact convergence in FL.

+ The adaptive initial model construction strategy, leveraging gradient similarity, is innovative. This method enables the model to adapt more rapidly to the currently active clients by aligning more closely with their data distributions, potentially enhancing training efficiency.

**Weaknesses:**

- The theoretical analysis relies on a strong convexity assumption, which does not typically apply to deep neural networks commonly used in FL. Since neural networks are often non-convex, this assumption limits the practical applicability of the derived bounds in FL scenarios involving complex models.

- Using a single pilot model to initialize each new round, while computationally efficient, may introduce bias if client participation patterns shift significantly or if clients with distinct data distributions join later in training. A single pilot model might not fully capture the diversity of evolving data distributions, potentially affecting the model’s generalizability across client types.

- Although the experimental evaluation is thorough, it focuses solely on image classification tasks with a relatively small client population (10 clients). Including tests on non-image data, such as the Shakespeare dataset, and expanding the number of clients would provide solid evidence of the framework’s scalability and versatility.

- Lastly, the framework does not explicitly address the issue of catastrophic forgetting, which may occur when clients rejoin after an absence. Adapting exclusively to active clients could result in the loss of knowledge from previously encountered data distributions, thereby reducing performance for returning clients, especially if their data significantly diverges from that of current participants.

**Questions:**

- Could the implications of the strong convexity assumption for practical deep learning scenarios be discussed in more detail? Additionally, are there ways the analysis could be extended to accommodate non-convex settings commonly found in FL?

- What potential strategies could be employed to mitigate the bias introduced by using a single pilot model? For instance, would periodically updating the pilot model or utilizing multiple pilot models help in capturing the diversity of evolving data distributions?

- How does the proposed framework perform under different types of data (such as the Shakespeare dataset) or more complex datasets (such as the Tiny ImageNet dataset)? Additionally, further experiment on increasing the number of clients is necessary to demonstrate the proposed method’s effectiveness.

- What strategies could be integrated into the framework to mitigate catastrophic forgetting? Additionally, has there been any analysis or experiments on how the method performs when clients with previously seen data distributions rejoin the training process?

---

> ### Author Response · Authors · 2024-11-21
> **Question 1 and Weakness 1**
>
> While our analysis is built upon the strong convexity assumption, this choice primarily serves to gain deeper insights into the dynamics of our system and to identify key factors that influence performance. The strong convexity assumption allows us to rigorously analyze how different data distributions (modeled by the probability distributions used to form local datasets) affect the deviation between optimal models. Leveraging these insights, we designed our initial model reconstruction algorithm to mitigate performance drops due to shifts in data distribution caused by client arrivals and departures.
>
> In more static federated learning scenarios where clients are fixed, and no new clients join or leave the system, the analysis can indeed be extended to non-convex loss functions. However, in more dynamic scenarios like ours—where the optimization goal changes from round to round due to client churn and evolving datasets—the strong convexity assumption becomes more critical for deriving meaningful theoretical insights. Notably, even in simpler dynamic settings where only one new client is allowed to join the system during training, previous work [4] still relies on the strong convexity assumption.
>
> That being said, we recognize the importance of extending our analysis to accommodate non-convex settings, which are more reflective of practical deep learning scenarios. Although our theoretical framework currently assumes strong convexity, our experiments conducted in non-convex settings (e.g., using deep neural networks) demonstrate similar trends in terms of performance and convergence. This empirical evidence suggests that our approach remains effective even when the strong convexity assumption does not hold, providing a promising direction for future work to generalize the theoretical analysis to non-convex cases.
>
> [4] Ruan, Yichen, et al. "Towards flexible device participation in federated learning." International Conference on Artificial Intelligence and Statistics. PMLR, 2021.

---

> ### Author Response · Authors · 2024-11-21
> **Question 2 and Weakness 2**
>
> Thank you for your insightful comment. However, we would like to clarify that the pilot model is not used as the initial model for every training round. Instead, we use the pilot model as an initial point for a few additional federated training rounds specifically to compute gradients when there is a shift in the data distribution. These gradients are designed to capture changes in the time-varying data distributions and play a crucial role in assessing the similarity between data distributions across different rounds. This similarity is then used to compute the weights assigned to past global models, which are aggregated to form the true initial model for each new round.
>
> If your concern is that we use only one pilot model to compute the gradients, our experiments have shown that, regardless of the data distributions on which the models constituting the pilot model were originally trained, the computed gradients effectively reflect the similarity between evolving data distributions. This ensures that our method adapts well to changing client data, thereby reducing potential bias in the initialization process.
>
> That said, we acknowledge your observation and agree that the model’s performance could be further enhanced. For instance, periodically updating the pilot model or using multiple pilot models to compute gradients could better capture the diversity of evolving data distributions. These strategies could help mitigate potential biases, especially in scenarios with significant shifts in client participation or when clients with distinct data distributions join later in training. Exploring these extensions is an exciting direction for future research to improve the generalizability of our approach.

---

> ### Author Response · Authors · 2024-11-21
> **Question 3 and Weakness 3**
>
> Thank you for raising your concern. We have conducted additional experiments for Tiny Imagenet dataset with 20 clients. The results are included in Table 2 on Page 24 in Appendix due to the space constraint.

---

> ### Author Response · Authors · 2024-11-21
> **Question 4 and Weakness 4**
>
> Thank you for indicating this potential issue. Actually, our proposed algorithm directly addresses this problem. Consider a simple example: in one round, we have three clients, A, B, and C. Clients A and B have classes 1 and 2, and client C has classes 5 and 6. The trained model for this round is denoted as $\mathbf{w}^{(1)}$. This model should perform well for clients A, B, and C. In the next round, clients A and B leave, and clients D and E join, with D and E having classes 3 and 4. The trained model for this round is denoted as $\mathbf{w}^{(2)}$. Then, in the following round, clients D and E leave, and clients A, B, and C rejoin. After applying the dynamic initial model construction, the constructed initial model would be based on $\mathbf{w}^{(1)}$, and this model should perform well for clients A, B, and C.
>
> Although a model specifically trained for classes 5 and 6 (the classes that client C has) may outperform $\mathbf{w}^{(1)}$ for those classes, creating a personalized model for each individual client is not the goal of federated learning (nor the focus of our work). The primary objective of federated learning is to maintain a global model that performs well across all clients. Our method’s dynamic initial model construction ensures that the global model adapts to the arbitrary joining and leaving of clients, as well as shifts in data distribution, thereby mitigating catastrophic forgetting. This enables the model to maintain consistent performance across all clients, even as the set of clients and their data distributions evolve, providing a robust and scalable solution to the problem.

---

> > ### Comment · Reviewer_jhbc · 2024-11-26
> >
> > The reliance on the strong convexity assumption provides a solid theoretical foundation for analyzing dynamic federated learning scenarios. However, this assumption restricts the framework's practical use in non-convex settings, which are more common in deep learning tasks within federated learning.
> >
> > The proposed dynamic initial model construction method helps reduce catastrophic forgetting by adapting the global model to both current and past clients. However, it mainly focuses on maintaining the global model’s performance and does not fully address the challenges of clients rejoining with different data distributions.
> >
> > For these reasons, I will maintain my current rating. Thank you for your efforts to improve the work.

---

### Official Review · Reviewer_nEFP · 2024-11-02

**Soundness:** 2
**Presentation:** 2
**Contribution:** 1
**Rating:** 3
**Confidence:** 3

**Summary:**

The paper tackles the problem of dynamic federated learning in which clients can join or leave during training. Unlike traditional FL approaches that assume a static set of clients, this paper addresses the challenges of an evolving client pool, which can disrupt convergence and slow adaptation. The authors propose a probabilistic framework to model client types and a dynamic optimization objective to maintain model relevance. They introduce an adaptive initial model construction strategy based on weighted averaging of gradients, prioritizing models aligned with current data.

**Strengths:**

1. The paper is well-written and easy to follow.

2. The authors provides the theoretical analysis, including convergence bounds and an upper bound on the optimality gap.

3. The proposed method is tested on diverse datasets, label distributoins.

**Weaknesses:**

1. I am uncertain about the novelty of this dynamic FL setting where clients join or leave during training. To me, this scenario seems fundamentally similar to federated learning with concept drift, as discussed in [1].

2. Although the theoretical analysis adds rigor, the proof relies on too many inequalities to approximate results, which makes the final theoretical outcomes feel quite loose. Does this theoretical analysis truly reflect the actual training dynamics?

3. The proposed Pilot Model requires storing all previous global models, which could lead to significant storage issues.

4. The experimental section is relatively weak; evaluating only FedAvg and FedProx does not sufficiently demonstrate the effectiveness of the proposed method. I think methods designed for FL with concept drift could also apply well to this dynamic FL setting.

[1] Federated Learning under Distributed Concept Drift

**Questions:**

Please see the weakness section.

---

> ### Author Response · Authors · 2024-11-20
> **Weakness 1**
>
> Thank you for your question. Although our work shares one common aspect with the aforementioned paper—the dynamic nature of the objective function—the problem formulation and the approach to solving it are entirely different. In our work, we focus on training a single model and constructing the initial model to adapt to shifts in the data distribution. In contrast, the referenced work trains multiple models to handle multiple concept drifts and uses similarities in data distributions to cluster clients into different groups.
>
> Moreover, we present two key innovations. First, we provide an upper bound on the optimality gap for this scenario, expressed in terms of the probability distributions governing local dataset formation and other learning parameters. Unlike previous approaches, which measure non-IIDness through the loss function, we are among the first to directly connect probability distributions with performance in a dynamic system, where client arrivals and departures follow arbitrary patterns, an analysis not provided in the referenced work. Second, we approach the problem differently by leveraging historical information about the data distribution to construct the initial model. This helps mitigate training loss in the presence of concept drift, which results from client arrivals and departures.
>
> When considering transient optimization goals across different global rounds, our proposed algorithm is particularly effective in providing a good initial point for reaching the optimal solution in each round, enabling rapid adaptation. This is especially useful in scenarios where client arrival and departure patterns are erratic, with clients potentially leaving after only a few global rounds.

---

> ### Author Response · Authors · 2024-11-21
> **Weakness 2**
>
> As seen in many papers, including recent articles on federated learning [1-3], convergence analysis for federated learning often requires approximations to derive meaningful bounds. In our analysis, we acknowledge that exact results are often difficult to obtain due to the inherent complexity of dynamic client participation and time-varying local datasets. Therefore, the use of approximations is a standard approach in this field, particularly when precise closed-form solutions are unavailable.
>
> We have made a conscious effort to ensure that the bounds we derive are as tight as possible within the context of the approximations we use. Specifically, we only resort to approximations when exact equality cannot be derived, and all approximations employed are based on well-established, standard techniques that have been widely used in the literature for similar types of analysis. These methods allow us to maintain a balance between theoretical rigor and practical applicability, ensuring that our results provide a meaningful and reasonable reflection of the underlying training dynamics.
>
> While it is true that the final theoretical outcomes might appear somewhat loose due to the use of inequalities, this is a common trade-off in convergence analysis, where the goal is to derive general, applicable results that hold under a wide range of conditions. The approximations we employ are necessary to account for the complex, dynamic nature of federated learning, and we believe that they still capture the essential aspects of the training dynamics. Additionally, we have conducted extensive experiments, and the results demonstrate that our approach achieves good performance, further validating the theoretical analysis.
>
> [1] Chen, Wenlin, Samuel Horváth, and Peter Richtárik. "Optimal Client Sampling for Federated Learning." Transactions on Machine Learning Research, 2022.
>
> [2] Wang, Shiqiang, and Mingyue Ji. "A Lightweight Method for Tackling Unknown Participation Statistics in Federated Averaging." The Twelfth International Conference on Learning Representations, 2024.
>
> [3] Wang, Yujia, et al. "FADAS: Towards Federated Adaptive Asynchronous Optimization." arXiv preprint arXiv:2407.18365 (2024).

---

> ### Author Response · Authors · 2024-11-21
> **Weakness 3**
>
> In both our proposed algorithm and simulation, the number of global models used to construct the pilot model is fully customizable based on the specific requirements of the scenario. This flexibility allows you to adjust the number of models depending on your storage capacity and performance needs. If storage is a concern, you can simply reduce the number of models used without significantly impacting the effectiveness of the pilot model.
>
> In our conducted experiments, we achieved good performance using no more than 3 global models to construct the pilot model, which kept the storage requirements minimal. Additionally, once the pilot model is created, the global models used in its construction can be discarded if storage space is limited. This design ensures that our method can adapt to various storage constraints while maintaining strong performance, allowing users to balance between resource availability and model accuracy as needed.

---

> ### Author Response · Authors · 2024-11-21
> **Weakness 4**
>
> Thank you for your feedback. In the updated paper, we also present simulation results for the advanced federated learning algorithm, Stochastic Controlled Averaging for Federated Learning (SCAFFOLD), in Table 1 on Page 9 (highlighted in blue). SCAFFOLD is designed to mitigate the client drift issue in federated learning by controlling local updates. It uses control variates to reduce the variance of these updates, resulting in more stable and efficient convergence across federated learning systems. Our findings are consistent with this approach, and we reach the same conclusion.

---

> ### Comment · Reviewer_nEFP · 2024-11-25
> **Thanks so much for author's reply.**
>
> I appreciate the authors' response to my comments. However, given the strong assumptions and the limited experimental contributions (even with the addition of comparison with SCAFFOLD (published in 2019), which itself is not sota), I am inclined to maintain my current score.

---

### Official Review · Reviewer_bynj · 2024-11-04

**Soundness:** 2
**Presentation:** 2
**Contribution:** 1
**Rating:** 3
**Confidence:** 4

**Summary:**

This paper addresses a federated learning (FL) scenario where clients can dynamically join or leave the system. The authors propose a dynamic optimization objective tailored to online(active) clients, incorporating a probabilistic model to aid in convergence analysis for the new global optimization objectives. Experiments are conducted across various settings to validate the proposed approach.

**Strengths:**

The authors devote considerable effort to the convergence analysis, ensuring that the divergence between the realistic output $w^{(g)}$ and the global optimal point $w^{(g)*}$ at each round $g$ is bounded. Through recursive iterations, they demonstrate that the divergence for each round remains within an upper bound, enhancing the reliability of their proposed method.

**Weaknesses:**

The novelty of the system design warrants scrutiny. In a typical horizontal FL framework, at any round $g$, it is both natural and reasonable to compute the global model using the current active (or online) clients. If we denote the set of active clients as ${1,2,...,C}$ and the final local model for a client $c \in {1,...,C}$ as $w^c$, we can compute the global model $w^g = \sum_{c \in {1,...,C}} \eta_c w^c$ with the constraint $\sum_{c \in {1,...,C}} \eta_c = 1$, following standard aggregation rules. In the examples provided, the weight for each client is based on the ratio of the client's data samples to the total data samples of online clients, as described on line 214. Given this, it is unclear why the authors introduce a “transitive global objective” $w^{(g)}$ tailored solely for the active client set at round $g$. In a realistic horizontal FL setting, **the server can only compute the weighted average over active clients**, regardless of whether the clients hold heterogeneous data distributions. Therefore, introducing a changing objective in the realistic experiment setting does not appear to provide substantial novelty or practical benefit.

Regarding the experimental results, the accuracy improvements in Table 1 seem attributable to the dynamic initial model construction described in Section 5. However, the pilot model design appears independent of the proposed dynamic objective and could be implemented in any realistic FL scenario, provided the aggregated model from each round $g$ is retained. My concern lies in understanding how the dynamic objective impacts the final performance. This connection remains unclear, and I believe the emphasis in Section 5 and the experimental results downplay the significance of the proposed dynamic objective. Can the authors clarify this aspect?

**Questions:**

1. **Purpose of the Dynamic Objective**: For further clarification on the motivation behind the dynamic global objective, please refer to the detailed comments in the *Weaknesses* section.
2. **Clarity of Equations**: The presentation of Equations 1 and 2 is difficult to follow due to excessive notation. A more straightforward formulation with minimal notation might improve readability and comprehension.
3. **Convergence Condition in Theorem 1**: To guarantee convergence, the coefficient component in $(a)$, specifically $\mu\eta^{(g)}\left(\sum_{k \in \mathbb{K}^{(g)}}\frac{D^{(g)}_k e^{(g)}_k}{D^{(g)}}\right)$, must be $< 2$. How is this condition ensured? It would be helpful if the authors could provide a synthetic experiment with a clearly defined $\mu$, $\eta^{(g)}$, and other relevant hyperparameters to demonstrate that the objective function indeed converges under these settings.

---

> ### Author Response · Authors · 2024-11-20
> **Question 1 and First Paragraph in Weaknesses**
>
> Thank you for your question, which provides an opportunity to clarify our work. What you described corresponds to federated learning with partial participation. In this line of research, authors typically aim to design a client sampling strategy (e.g., [1]) or modify the aggregation weight $\eta_c$ (e.g.,[2]) to minimize a static global loss function that remains constant over time. As seen in Equation 1 of both [1] and [2], the global loss function is defined as a weighted sum of the local loss functions of a predefined set of $N$ clients: $\sum_{i=1}^{N} \eta_i f_i(x)$, where $f_i(x)$ is the local loss function of client $i$ and $\eta_i$ is the corresponding aggregation weight. Note that here $N \geq C$ because not all clients participate in training during each round, and the set of $N$ clients must be known beforehand to define the global loss function. Thus, the optimization objective is static, which aligns with the traditional partial participation model illustrated in Figure 1 of our paper.
>
> To be more specific, in a typical horizontal federated learning (FL) setting, the server can only compute a weighted average over the active clients, but the objective remains to obtain a model that minimizes $\sum_{i=1}^{N} \eta_i f_i(x)$, not $\sum_{i=1}^{C} \eta_i f_i(x)$. This approach assumes prior knowledge of all clients that may potentially participate throughout the training process, corresponding to a predefined set of $N$ clients. In contrast, the more dynamic setting we consider does not assume any prior knowledge of which clients may join or leave the system during its lifecycle. This is the key distinction between our system model and existing approaches. Given this dynamic environment, it is more reasonable to consider a transient global objective that adapts to the currently active clients.
>
> In summary, if one assumes that the server can only compute the weighted average over active clients to obtain a model $w^g = \sum_{c=1}^{C} \eta_c w^c$ that optimizes $\sum_{i=1}^{N} \eta_i f_i(x)$, then this scenario falls under federated learning with partial participation—a well-studied area. However, if one instead assumes that the server computes the weighted average over active clients to optimize a transient global loss function $\sum_{i=1}^{C} \eta_i f_i(x)$, this is precisely the setting we address. We not only propose a new theoretical framework but also develop a novel algorithm for this dynamic environment. We hope this clarifies the novelty of our work. We have updated the ‘Illustrative Examples’ paragraph on Page 2 to further emphasize the key differences.
>
>
> [1] Chen, Wenlin, Samuel Horváth, and Peter Richtárik. "Optimal Client Sampling for Federated Learning." Transactions on Machine Learning Research, 2022.
>
> [2] Wang, Shiqiang, and Mingyue Ji. "A Lightweight Method for Tackling Unknown Participation Statistics in Federated Averaging." The Twelfth International Conference on Learning Representations, 2024.

---

> ### Author Response · Authors · 2024-11-20
> **Second Paragraph in Weakness**
>
> Thank you for your insightful question. The use of the pilot model is essential for dynamically constructing the initial model in our approach. Rather than relying on a randomly initialized global model, we use the pilot model as the starting point for several global rounds, which enables us to compute the gradients more effectively. These gradients capture the evolving data distributions and are used to assess the similarity between the data distributions of different rounds. By comparing these gradient similarities, we can compute weights that help construct the initial model for subsequent rounds.
>
> Moreover, it is possible to periodically update the pilot model to better keep up with shifts in the data distribution, which could lead to further improvements in performance. This flexibility provides great potential for additional gains. However, in our experiments, we found that even when using the same pilot model throughout the rounds, the gradients still effectively capture the similarities between the evolving data distributions. This highlights the robustness of our method in handling data shifts without the need for frequent updates to the pilot model.
>
> In the following two paragraphs, we aim to illustrate how the dynamic initial model construction is tied to the dynamic objective function. To enhance readability, we will continue using the notation from Question 1. At global round $g_1$, suppose we have active clients indexed from $1, \dots, C_1$, with a transient global loss function given by $\sum_{i=1}^{C_1} \eta_i f_i(x)$. At global round $g_2$, active clients are indexed from $1, \dots, C_2$, with a different transient global loss function $\sum_{i=1}^{C_2} \eta_i f_i(x)$. If the data distribution at round $g_1$ is similar to that at round $g_2$ (with $g_2 > g_1$), the model that minimizes the loss at round $g_1$ should also be similar to the model that minimizes the loss at round $g_2$. In our dynamic initial model construction, the model constructed for round $g_2$ will be closely aligned with the model trained in round $g_1$, providing an effective starting point and enabling fast adaptation in round $g_2$.
>
> However, if the optimization goal is to find a model that minimizes the static global loss function $\sum_{i=1}^{N} \eta_i f_i(x)$, then using our approach to construct an initial model would not be meaningful, as it could be far removed from the model that minimizes the static objective. Therefore, the pilot model design, which is an integral part of our proposed algorithm, is tightly connected to the dynamic global objective we aim to optimize.
>
> Not only is the proposed algorithm and its experimental results closely linked to the dynamic objective, but our convergence analysis also reflects this in both Lemma 1 and Theorem 1. Since we are considering a dynamic objective, the optimal global model varies across rounds. In Lemma 1, we quantify the deviation between two optimal global models by considering the characteristics of each global round (i.e., the data distribution of each client's local dataset, the size of the local datasets, and the total number of clients per round). If we were to consider a static objective, the deviation would be zero. Additionally, because we are dealing with a dynamic objective, the term that needs to be bounded in Theorem 1 is the round-specific optimality gap between the global model $\mathbf{w}^{(g)}$ and the optimal model $\mathbf{w}^{(g)\ast}$ for that particular round. If we were instead considering a static objective, the optimality gap would be defined as the difference between the global model $\mathbf{w}^{(g)}$ and the static optimal model $\mathbf{w}^{\ast}$.

---

> ### Author Response · Authors · 2024-11-20
> **Question 2**
>
> Thank you for your feedback. In our updated manuscript, we have significantly simplified the notations for the definitions of global and local loss functions on Page 4, as well as in the proof of Lemma 1 on Page 14.

---

> ### Author Response · Authors · 2024-11-21
> **Question 3**
>
> Thank you for your insightful comment. In theory, given the strong convexity constant, the number of participating clients, and the size of each client's local dataset in a given round, we can control the learning rate $\eta^{(g)}$ and the number of local SGD iterations to ensure that $\mu \eta^{(g)}\left(\sum_{k \in \mathbb{K}^{(g)}} \frac{D_k^{(g)}e_k^{(g)}}{D^{(g)}}\right) < 2$. This is a standard practice in optimization-based algorithms, where careful tuning of the learning rate and the number of iterations can easily satisfy such conditions.
>
> Moreover, it is conventional in the federated learning literature to derive theoretical upper bounds on the convergence performance, demonstrating that the algorithm has the potential to converge under properly selected hyperparameters. These theoretical guarantees are often sufficient to validate the feasibility of the convergence condition, as they provide a clear framework for ensuring the condition holds without the need for extensive experimental verification. We have stated the condition for convergence explicitly after presenting our main theoretical result in Theorem 1 on Page 6 when discussing term (a).

---

> ### Comment · Reviewer_bynj · 2024-11-26
>
> I appreciate authors' efforts to address my concerns.
>
> While the authors have outlined the distinction between FL with partial participation and their proposed setting, I still find it challenging to fully grasp the practical difference between the two. In my understanding, both scenarios compute a weighted average of the models uploaded by active clients in each round, and the transient global objective does not seem to significantly affect the aggregation process.
>
> I now have a better understanding of the connection between the pilot model and the proposed objective, thanks to the authors' explanation. However, this linkage was not adequately clarified in the initial version. I encourage the authors to pay closer attention to the logical flow between sections, as this is especially important when submitting to a top-tier conference.
>
> The convexity proof relies on assumptions that I believe are somewhat restrictive and may require further refinement.
>
> Given these points, I am inclined to keep my original score.

---

### Meta-Review · Area_Chair_rgkC · 2024-12-17

**Metareview:**

The paper studies federated learning under dynamic client participation. Main contributions include a bound on the optimality gap in terms of the gradient noise, local training iterations, data heterogeneity, and dynamic client participation, as well as an adaptive scheme using weighted averaging guided by client similarity.

Strengths include the derivation of the above bounds. Reviewers remained concerned about the strength of the assumptions used, but also about the experiment section that considered only quite old competitors.

**Additional Comments On Reviewer Discussion:**

Reviewers remained concerned about the presentation clarity, the relationship to traditional FL with partial participation, the non-SoTA competitors in experients, the strength of assumptions used, the treatment of client heterogeneity, and asked for several improvements, particularly in baselines used.

---

### Decision · Program_Chairs · 2025-01-22

Reject